# PIM kinase control of CD8 T cell protein synthesis and cell trafficking

Julia M Marchingo[1]*[†], Laura Spinelli[1,2], Shalini Pathak[1‡], Doreen A Cantrell[1]*

[1]Cell Signalling and Immunology Division, School of Life Sciences, University of Dundee, Dundee, United Kingdom; [2]Molecular Cell and Developmental Biology Division, School of Life Sciences, University of Dundee, Dundee, United Kingdom

*For correspondence:
marchingo@wehi.edu.au (JMM);
d.a.cantrell@dundee.ac.uk (DAC)

Present address: [†]Walter and Eliza Hall Institute of Medical Research, Melbourne, Australia; [‡]Innovation Factory Core Technology Facility, University of Manchester, Manchester, United Kingdom

Competing interest: The authors declare that no competing interests exist.

## eLife Assessment

These **important** findings detail the role of Pim1 and Pim2 in controlling the behaviour and activity of 'killer' T cells; a vital cell within of our immune system. The authors capitalized on high resolution quantitative analysis of the proteomes and transcriptomes of Pim1/Pim2-deficient CD8 T cells to provide **compelling** evidence for how the PIM1/2 kinases control TCR-driven activation and IL-2/IL-15-driven proliferation and differentiation into effector T cells. It's also noteworthy that Pim1/Pim2 impact is better revealed through quantitative proteomics than transcriptomics.

**Abstract** Integration of kinase signalling networks co-ordinates the transcriptional, translational, and metabolic changes required for T cell activation and differentiation. This study explores the role of the Serine/Threonine kinases PIM1 and PIM2 in controlling mouse CD8 T lymphocyte antigen receptor-mediated activation and differentiation in response to the cytokines Interleukin-2 (IL-2) or IL-15. We show that the PIM kinases are dispensable for antigen-receptor and IL-15 controlled differentiation programs, but that they play a selective role in IL-2 regulated CD8 T cell fate. One key insight was that PIM kinases controlled the migratory capabilities of effector CD8 T cells, with *Pim1/Pim2*-deficient CD8 T cells unable to fully switch off the naive T cell chemokine and adhesion receptor program during effector differentiation. PIM kinases were also needed for IL-2 to sustain high expression of the glucose transporters SLC2A1 and SLC2A3 and to maintain activity of the nutrient-sensing kinase mTORc1. Strikingly, PIM kinases did not have a dominant impact on IL-2-driven transcriptional programs but rather selectively modulated protein synthesis to shape cytotoxic T cell proteomes. This study reveals a selective role of PIM kinases in IL-2 control of CD8 T cells and highlights how regulated changes in protein synthesis can impact T cell phenotypes.

## Introduction

CD8 T lymphocytes are key effector cells for adaptive immune responses to viruses, intracellular pathogens, and tumours. Their clonal expansion and differentiation to cytotoxic effector or memory CD8 T cells is controlled by signalling pathways initiated by the T cell antigen receptor (TCR), co-stimulatory molecules, and cytokines. Receptor engagement by immune stimuli triggers the activation of cytosolic tyrosine kinases that then direct the activity of a diverse network of Serine/Threonine kinases, disseminating signals from plasma membrane to cell interior. The Src family and ZAP70 Tyrosine kinases initiate this process downstream of the TCR, with the Janus kinases (JAK) as the main signal transducer for important cytokine regulators of T cell differentiation, IL-2 and IL-15 (**Cantrell, 2015**; **Saravia et al., 2020**). The subsequent network of Serine/Threonine kinases that is engaged beyond these initiating events coordinates phosphorylation of a diverse array of protein targets. These protein targets include chromatin regulators, transcription factors, translation machinery, cytoskeletal proteins, and

metabolic enzymes, which then control the changes in transcriptional, translational, and metabolic programs that drive effector and memory T cell differentiation (*Navarro et al., 2011*; *Tan et al., 2017*; *Locard-Paulet et al., 2020*).

CD8 T cells express ~200 Serine/Threonine protein kinases (*Navarro et al., 2014*; *Howden et al., 2019*). There is a good understanding of the roles of some Serine/Threonine kinases in CD8 T cells, including the role of the diacylglycerol regulated kinases, MAP kinases ERK1 and 2, Phosphatidylinositol (3,4,5) triphosphate- controlled kinases PDK1 and AKT, and the nutrient-sensing kinases mammalian Target of Rapamycin complex 1 (mTORc1) and AMP-activated protein kinase (AMPK; *Finlay et al., 2012*; *Rolf et al., 2013*; *Blagih et al., 2015*; *Cantrell, 2015*; *Hukelmann et al., 2016*; *Tan et al., 2017*; *Howden et al., 2019*; *Saravia et al., 2020*; *Damasio et al., 2021*; *Spinelli et al., 2021*). However, beyond these core kinases there is much still to learn about how many of the ~200 Serine/Threonine kinases expressed in CD8 T cells contribute to the control of T cell function. In this context, the PIM kinases are a family of Serine/Threonine kinases that are strongly and rapidly induced in T cells responding to immune stimuli (*Aho et al., 2005*; *Fox et al., 2005*; *Peperzak et al., 2010*; *Jackson et al., 2012*). The PIM family contains three Serine/Threonine kinases, PIM1, PIM2, and PIM3, of which T lymphocytes predominantly express PIM1 and PIM2 (*Mikkers et al., 2004*; *Fox et al., 2005*). As PIM kinases are constitutively active, their participation in signalling cascades is dictated by regulated changes in their protein expression levels (*Warfel and Kraft, 2015*). PIM1 and PIM2 have little to no expression in naïve T cells but are induced by T-cell receptor activation and costimulation (e.g. CD27) (*Wingett et al., 1996*; *Fox et al., 2005*; *Peperzak et al., 2010*). PIM kinases are also direct transcriptional targets of JAK/STAT signalling pathways and as such PIM1 and/or PIM2 are rapidly induced in T cells downstream of a diverse array of cytokines, including multiple common cytokine receptor γ-chain family cytokines, IL-6 and IL-12 (*Matikainen et al., 1999*; *Aho et al., 2005*; *Fox et al., 2005*; *Warfel and Kraft, 2015*; *Buchacher et al., 2023*).

There have been some analyses of how deletion of PIM kinases impacts CD8 T cell function. A mild reduction in *Pim1/Pim2/Pim3*-deficient CD4 and CD8 T cell proliferation has been observed in response to sub-optimal TCR stimulation and IL-2 (*Mikkers et al., 2004*), and there is evidence that PIM1 and PIM2 control survival pathways in naïve CD4 and CD8 T cells in concert with mTORc1 (*Fox et al., 2005*). *Pim1/Pim2/Pim3* deletion is reported to promote expression of stemness and memory-associated transcription factors including Bcl6 and Tcf7 (*Chatterjee et al., 2019*), and PIM kinase activity has been linked with maintenance of memory CD8 T cell survival (*Knudson et al., 2017*). It has also been shown recently that *Pim1/Pim2* deficiency prevented mTORc1 activation and the proliferation of non-conventional intra-epithelial gut CD8 T lymphocytes (IEL) in response to the cytokine IL-15 (*James et al., 2021*).

Elevated PIM kinase expression has been noted in patient T cells for a number of autoimmune conditions, including rheumatoid arthritis, uveitis and coeliac disease, and PIM kinase inhibition has been proposed as a possible intervention to dampen autoimmunity (*James et al., 2021*; *Maney et al., 2021*; *Li et al., 2022*). PIM kinase inhibitors have also entered clinical trials to treat some cancers (e.g. multiple myeloma, acute myeloid leukaemia, prostate cancer), and although they have not been effective as a monotherapy, there is interest in combining these with immunotherapies. This is due to studies showing PIM inhibition reducing expression of inhibitory molecules (e.g. PD-L1) on tumour cells and macrophages in the tumour microenvironment and a reported increase of stem-like properties in PIM-deficient T cells which could potentially drive longer lasting anti-cancer responses (*Chatterjee et al., 2019*; *Xin et al., 2021*; *Clements and Warfel, 2022*). However, PIM kinase inhibition has also generally been shown to be inhibitory for T cell activation, proliferation, and effector activities (*Fox et al., 2003*; *Mikkers et al., 2004*; *Jackson et al., 2012*) and use of PIM kinase inhibitors could have the side effect of diminishing the anti-tumour T cell response. Given the role of CD8 T cells in anti-cancer immunity, it is important to understand how these kinases control CD8 T cell function. However, there has not yet been a systematic analysis of the role of PIM kinases in conventional CD8 T cells as they activate and differentiate to form effector and memory cells. Accordingly, the current study, uses high-resolution quantitative analysis of *Pim1/Pim2*-deficient CD8 T cell proteomes and transcriptomes to obtain a comprehensive and unbiased understanding of how PIM kinases control CD8 T cell TCR-driven activation, and IL-2 or IL-15-driven proliferation, and differentiation. The study reveals a selective role for PIM kinases as mediators of IL-2 signalling to control CD8 T cell trafficking and effector functionality. We uncover that the dominant impact of PIM kinase loss was

to shape effector CD8 T cell proteomes rather than transcriptomes, reflecting PIM kinase modulation of protein synthesis.

## Results

### PIM1 and PIM2 are strongly induced by antigen receptor engagement but are dispensable for T cell activation

T cell antigen receptor (TCR)-driven activation of naive T cells initiates their clonal expansion and differentiation into effector and memory cells. Quantitative mass spectrometry-based proteomics analysis of naive OT1 CD8 T cell stimulated with their cognate peptide over time (*Marchingo et al., 2020*) showed that PIM1 protein was strongly induced within 3 hr of TCR engagement, which was sustained at a 24-hr time point (*Figure 1A*). PIM2 expression was not detected until 9 hr post-stimulation but was also sustained at the 24-hr time point. 24-hr polyclonal activation of CD8 T cells with αCD3/αCD28 agonist antibodies similarly induced expression of PIM1 and PIM2 proteins (*Figure 1B*, *Supplementary file 1*). These increases in PIM kinase expression at a protein level were underpinned by transcriptional reprograming, with both *Pim1* and P*im2* mRNA very lowly expressed in naive CD8 T cells and strongly upregulated by immune activation (*Figure 1C*; *Spinelli et al., 2021*). Consistent with previous reports of Pim1 and Pim2 being the major PIM kinases in T cells (*Mikkers et al., 2004*; *Fox et al., 2005*), *Pim3* mRNA was only expressed at very low levels in naive and activated CD8 T cells (*Figure 1C*), with PIM3 protein not detected at all by mass spectrometry (*Supplementary file 1*).

PIM1 and PIM2 have similar substrate selectivity and hence functional redundancy (*Mikkers et al., 2004*; *Bullock et al., 2005*; *Fox et al., 2005*; *Peng et al., 2007*; *Warfel and Kraft, 2015*). Accordingly, to explore the role of the PIM kinases in T cells, mice with a whole body deletion of *Pim1* and *Pim2*, Pim1$^{-/-}$ (Pim1 KO) (*Laird et al., 1993*) and Pim2$^{-/-}$ or Pim2$^{-/Y}$ (Pim2 KO) (*Mikkers et al., 2004*), were backcrossed for >10 generations onto a C57BL/6 background and inter-crossed to generate *Pim1*/*Pim2* double knockout (Pim dKO) mice. There were normal proportions of peripheral T cells in spleens of Pim dKO mice (*Figure 1—figure supplement 1A*) similar to what has been reported previously in Pim dKO mice on an FVB/N genetic background (*Mikkers et al., 2004*), although the total number of T cells and splenocytes was lower than in age/sex matched wild-type (WT) mouse spleens (*Figure 1—figure supplement 1B–C*). This was not attributable to any one cell type (*Figure 1—figure supplement 1A*; *James et al., 2021*) but was instead likely the result of these mice being smaller in size, a phenotype that has previously been reported in *Pim1/2/3* triple KO mice (*Mikkers et al., 2004*).

We tested how Pim dKO T cells respond to immune activation and observed a normal increase in cell size in response to in vitro immune activation with αCD3/αCD28 agonist antibodies (*Figure 1D*). 24 hr αCD3/αCD28 activated Pim dKO CD4 and CD8 T cells also normally upregulated expression of established markers of T cell activation including CD25, CD44 and the transferrin receptor CD71 (*Figure 1E*). 24 hr TCR-activated Pim dKO CD8 T cells also produced normal levels of the key effector cytokine IFNγ (*Figure 1E*) and there was no discernible difference in TCR-driven proliferation between WT and Pim dKO T cells over 3 days in culture (*Figure 1F and J*). To systematically explore in an unbiased way if *Pim1*/*Pim2*-deficiency impacted TCR-driven activation phenotypes, we performed high-resolution mass spectrometry to measure the proteomes of 24 hr αCD3/αCD28 activated WT and Pim dKO CD4 and CD8 T cells. Proteomics analysis confirmed that no catalytically active PIM1 and PIM2 protein were made in Pim dKO mice (*Figure 1—figure supplement 2*). These experiments quantified expression of >7000 proteins but found no substantial quantitative or qualitative differences in protein content or proteome composition in activated WT versus Pim dKO CD4 and CD8 T cells (*Figure 1G–H*) (*Supplementary file 1*). Collectively, these results indicate that PIM kinases do not play an important unique role in the signalling pathways used by the TCR and CD28 to control T cell activation.

One prominent study has reported that *Pim1*/*Pim2* deficiency sensitises T cells to the mTORc1 inhibitor rapamycin (*Fox et al., 2005*), which suggests mTORc1 can substitute for the PIM kinases to control T cell activation. This previous study showed Pim dKO, but not WT, naive T cells cultured in IL-7 died in response to rapamycin treatment. It was also shown that rapamycin synergised with *Pim1*/*Pim2*-deficiency to substantially reduce TCR-driven activation and proliferation (*Fox et al., 2005*). However, we found no impact of *Pim1*/*Pim2*-deficiency on the ability of naive CD8 T cells cultured with IL-7 to survive in response to rapamycin treatment (*Figure 1I*). Furthermore, we found no difference in the

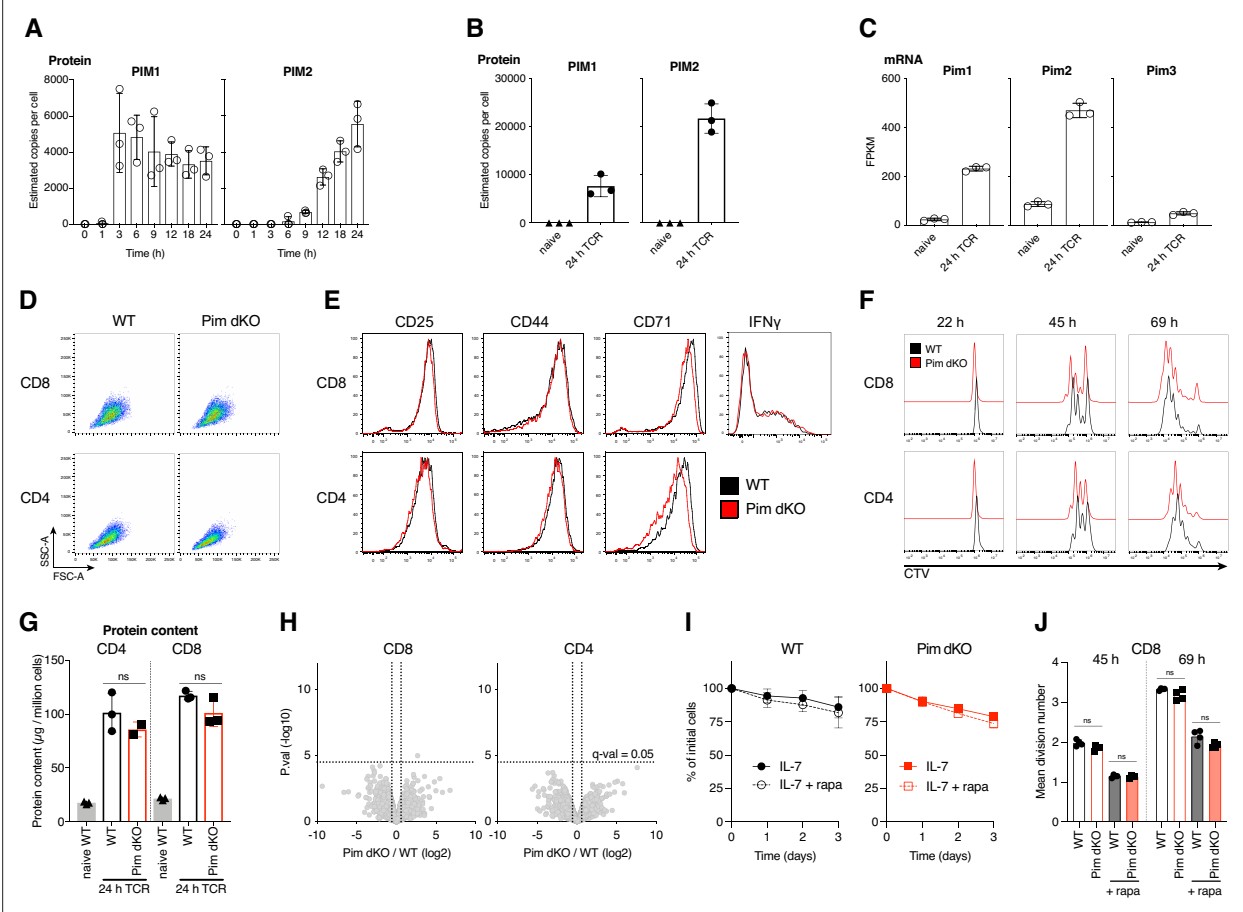

**Figure 1.** Pim1 and Pim2 are strongly TCR-induced, but dispensable for T cell activation. Estimated copies per cell of PIM1 and PIM2 protein from quantitative proteomics analysis of (**A**) OT1 CD8 T cells stimulated with SIINFEKL peptide for indicated times from published dataset (**Marchingo et al., 2020**) or (**B**) naive ex vivo and 24 hr αCD3/αCD28 (TCR) activated WT CD8 T cells (see (**G, H**)) for further details. (**C**) Fragments per kilobase million (FPKM) of *Pim1*, *Pim2,* and *Pim3* mRNA from published (**Spinelli et al., 2021**) bulk RNAseq analysis of naive and 24 hr gp33-41 peptide stimulated P14 CD8 T cells. Lymph node cell suspensions from C57BL/6 (WT) and *Pim1*KO/*Pim2*KO (Pim dKO) mice were activated for 24 hr with αCD3/αCD28 (both 0.5 µg/mL) and CD4 and CD8 T cell (**D**) FSC-A SSC-A profiles, (**E**) expression of surface activation markers (CD25, CD44, CD71) or CD8 T cell intracellular IFNγ were measured by flow cytometry. (**F**) Lymph node single-cell suspensions from WT and Pim dKO mice were labelled with CellTrace Violet (CTV), activated with αCD3/αCD28 (both 0.5 µg/mL) and CD4 and CD8 T cell CTV proliferation profiles were measured at indicated time points. (**G, H**) Lymph node cell suspensions from WT and Pim dKO mice were stimulated for 24 hr with αCD3/αCD28 (both 0.5 µg/mL) and activated CD4 and CD8 T cells were sorted for analysis by quantitative proteomics. Data was analysed using proteomic ruler method (**Wiśniewski et al., 2014**) to estimate protein copy number per cell. An interactive version of the proteomics expression data is available for exploration on the Immunological Proteome Resource website: immpres.co.uk (**G**) Total protein content (µg/million cells) (one-way ANOVA), (**H**) Volcano plots of p-value (-log₁₀) versus fold-change (log₂) in protein copy number between Pim dKO and WT. Horizontal dotted line represents multi-test correction cut-off of q=0.05, vertical dotted line shows 1.5-fold change. Phosphoribosyl Pyrophosphate synthase 1 like 1 (Prps1l1), was found to be higher in Pim dKO CD8 T cells, but was a low confidence quantification (based on only two unique peptides) with no known function in T cells. Lymph node single-cell suspensions from WT and Pim dKO mice were labelled with CellTrace Violet (CTV) and (**I**) cells were cultured in IL-7 (5 ng/mL) +/- rapamycin (20 nM) and CD8 T cell numbers measured over time or (**J**) cells were activated with αCD3/αCD28 (both 0.5 µg/mL) +/- rapamycin (20 nM) and CD8 T cell mean division number was calculated over time (two-way ANOVA). Symbols in bar charts represent biological replicates, symbols in (**I**) represent the mean. Error bars show mean ± S.D. Flow cytometry dot plots and histograms are representative of (**D, E**) n=3, except for IFNγ staining which is n=2, (**F**) n=5, or show pooled data from (**I**) n=3–4 and (**J**) n=5 biological replicates, with data collected over at least two independent experiments. Quantitative proteomics was performed on biological triplicates.

The online version of this article includes the following source data and figure supplement(s) for figure 1:

**Source data 1.** Raw values plotted in *Figure 1*.

**Figure supplement 1.** T cell counts in WT vs Pim dKO spleen.

**Figure supplement 1—source data 1.** Raw values plotted in *Figure 1—figure supplement 1*.

**Figure supplement 2.** Proteomics data confirms deletion of catalytically active PIM1 and PIM2.

**Figure supplement 2—source data 1.** Raw values plotted in *Figure 1—figure supplement 2*.

effect of rapamycin on mean division number in αCD3/αCD28 activated Pim dKO versus WT CD8 T cells (*Figure 1J*). Collectively, these data show that PIM 1 and PIM2 are rapidly and strongly expressed in response to immune activation, but they are dispensable for the antigen receptor and CD28-driven proteome remodelling that initiates T cell clonal expansion and differentiation. We also found no evidence that mTORc1 was substituting for *Pim1* and *Pim2* loss.

## PIM kinases do not control IL-15-driven CD8 T cell memory differentiation

The proliferative expansion and differentiation programs of antigen-activated CD8 T cells are shaped by cytokines. In this context, PIM1 and PIM2 kinase expression is regulated by the cytokines IL-2 and IL-15 which are key to shaping CD8 T cell effector and memory cell differentiation, respectively (*Weninger et al., 2001*; *Cornish et al., 2006*; *Buck et al., 2016*). Both IL-2 and IL-15 upregulate PIM1 and PIM2 expression in antigen primed CD8 T cells (*Figure 2A*) and are required to sustain PIM1 and PIM2 expression (*Rollings et al., 2018*). Accordingly, we assessed the contribution of PIM1 and PIM2 kinases to cytokine-driven expansion and differentiation of antigen activated CD8 T cells. In these experiments, antigen-primed CD8 T cells are expanded in IL-15 to generate memory phenotype CD8 T cells or expanded in IL-2 to generate effector cytotoxic T lymphocytes (CTL; *Weninger et al., 2001*; *Cornish et al., 2006*; *Buck et al., 2016*; *Figure 2B*).

Previous work has shown that PIM1 and PIM2 mediate IL-15 induced proliferation of gut intra-epithelial lymphocytes (IEL; *James et al., 2021*). However, their role in IL-15 responses in conventional antigen receptor activated CD8 T cells is not known. We therefore first examined the impact of *Pim1/Pim2*-deficiency on IL-15-driven conventional CD8 T cell expansion. Antigen experienced WT CD8 T cells show a robust proliferative response to IL-15 such that there is an ~90-fold expansion of the CD8 T cells between days 2 and 6 of culture (*Figure 2C*). Immune activated Pim dKO CD8 T cells also proliferated strongly in response to IL-15, but showed a moderate proliferative disadvantage resulting in an ~4-fold difference in total cell number by day 6 of the culture (*Figure 2C*). There was no impact of *Pim1/Pim2*-deficiency on the ability of IL-15 to sustain CD8 T cell viability (*Figure 2D*). Antigen-primed CD8 T cells expand in IL-15 to generate memory phenotype CD8 T cells (*Weninger et al., 2001*). We therefore examined whether *Pim1/Pim2*-deficiency had any impact on the ability of IL-15 to induce memory differentiation. Accordingly, we performed parallel bulk RNAseq and high-resolution mass spectrometry to analyse the transcriptome and proteome of IL-15 expanded WT and Pim dKO CD8 T cells. The RNAseq analysis quantified ~14,000 unique polyA+ RNA and using a cut off of >1.5 fold-change and q-value <0.05 we saw that the abundance of 381 polyA+ RNA was modified by *Pim1/Pim2*-deficiency (*Figure 2E*) (*Supplementary file 2A*). Of the 381 RNA that were detected as differentially expressed (FC >1.5, q<0.05) only ~30% were detected with an average expression over 10 transcripts per million (TPM), of these gene ~36% were non-coding RNA, uncharacterised genes or pseudogenes, leaving only 72 substantially expressed protein coding mRNA that were differentially expressed in Pim dKO versus WT IL-15 expanded CD8 T cells (*Supplementary file 2B*). These data indicate that Pim1/Pim2-deficiency has very little effect on IL-15 induced protein-coding mRNA expression in differentiated T cells. We then specifically examined if Pim dKO T cells had any differences in mRNA that are known to be critical for memory cell differentiation. We found that *Pim1/Pim2*-deficiency did not decrease expression of mRNA levels for critical memory T cell genes such as the adhesion molecules and chemokine receptors, *Sell* (CD62L), *Ccr7* and *S1pr1* that direct the trafficking of CD8 T cells to secondary lymphoid tissue. (*Figure 2F*). *Pim1/Pim2*-deficiency also did not reduce expression of mRNA for the key memory cell transcription factors *Tcf7, Foxo1, Foxo3 Klf2,* and *Id3*. (*Figure 2G*). Indeed, if anything, there were small increases in expression of the *Klf2, Id3,* and *Ccr7* mRNA in Pim dKO versus WT IL-15 expanded CD8 T cells (*Figure 2F and G*).

Critically, when the proteomes of IL-15 expanded WT and Pim dKO CD8 T cells were analysed and expression of ~6900 total proteins quantified, there were no major changes in the proteomes between WT and Pim dKO IL-15 expanded CD8 T cells (*Figure 2H*, *Supplementary file 3*). Of note, the mitochondrial proteome composition of IL-15 expanded WT and Pim dKO CD8 T cells were indistinguishable (*Figure 2I and J*). This is pertinent because one key feature of how IL-15 controls memory T cells is via mitochondrial remodelling to support energy production via oxidative phosphorylation and fatty acid oxidation (*Buck et al., 2016*), and it has been described that PIM kinases can control mitochondrial phenotypes in cancer cell lines and cardiomyocytes (*Din et al., 2013*; *Chauhan et al.,*

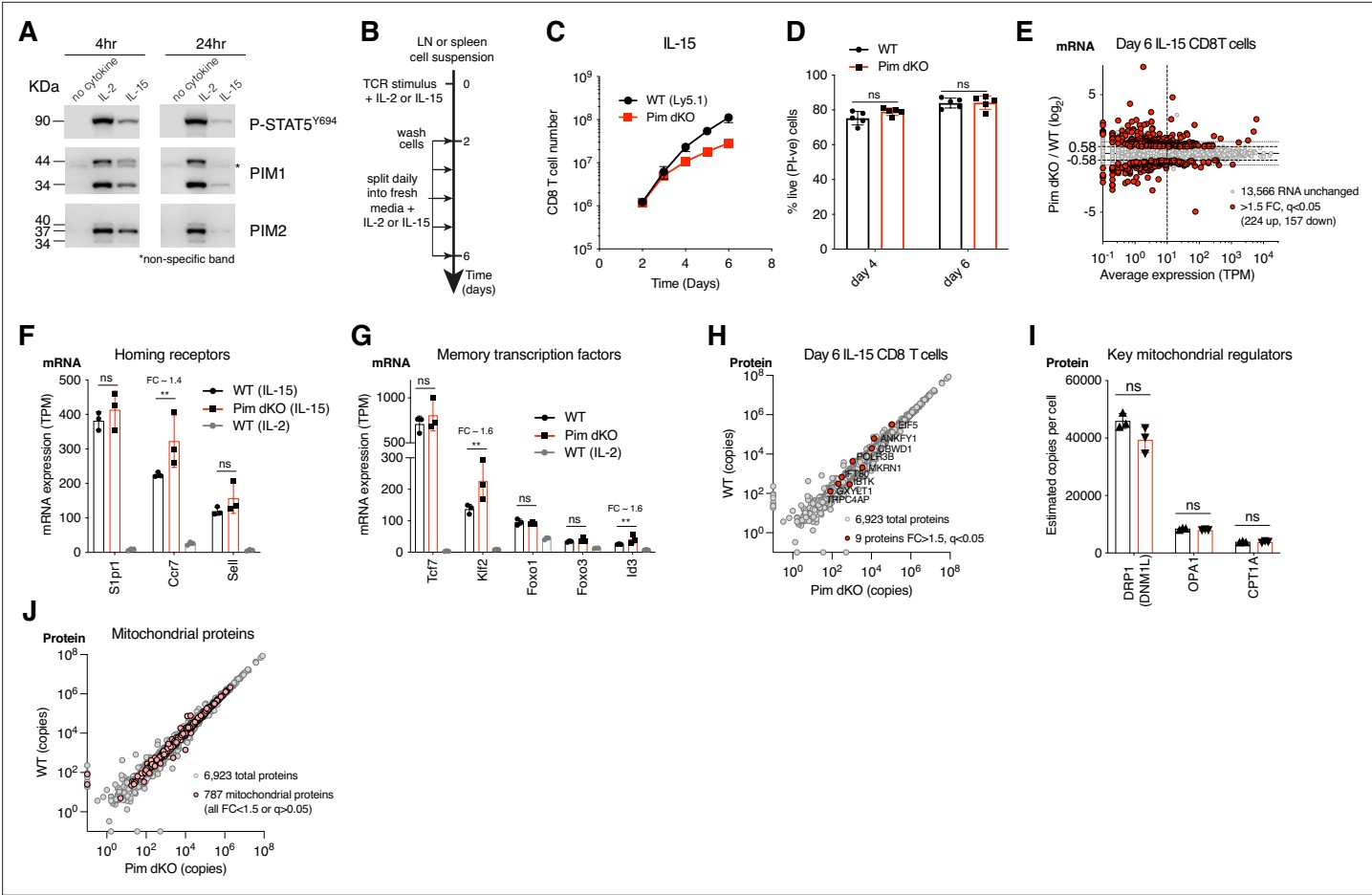

**Figure 2.** Pim1/Pim2 deficiency reduces IL-15-driven CD8 T cell proliferation but does not prevent memory differentiation. (**A**) OT1 lymph node cell suspensions were SIINFEKL peptide activated for 36 hr, washed then cultured with no cytokine, IL-15 (20 ng/mL) or IL-2 (20 ng/mL) for 4 or 24 hr. Western blots of PIM1 (two isoforms of 44 and 34 kDa, non-specific band indicated by *), PIM2 (three isoforms of 40, 37, and 34 kDa) or pSTAT5 Y694 expression. (**B**) Schematic of cytokine driven memory and effector CD8 T cell expansion and differentiation cultures. Lymph node or spleen cell suspensions were activated for 2 days with TCR stimulus + cytokine, washed, then split daily into fresh media + cytokine. (**C**) WT (Ly5.1) and Pim dKO LN suspensions were mixed at a 50:50 ratio for T cells and cultured as outlined in (**B**) with TCR stimulus αCD3/αCD28 (both 0.5 μg/mL) + cytokine IL-15 (20 ng/mL), and CD8 T cell number was measured daily. (**D**) WT and Pim dKO T cells were expanded with IL-15 in separate cultures as per (**B, C**) and % live cells (PI-ve) were assessed on days 4 and 6 (two-way ANOVA). (**E–J**) WT and Pim dKO CD8 T cells were activated with TCR stimulus αCD3/αCD28 (both 0.5 μg/mL) + cytokine IL-15 (20 ng/mL), expanded with IL-15 as per (**B**), with an additional CD4 T cell magnetic depletion step on day 3 of culture. CD8 T cells were harvested on day 6 for parallel RNAseq and proteomic analysis. An interactive version of the proteomics expression data is available for exploration on the Immunological Proteome Resource website: immpres.co.uk (**E**) Fold-change in mRNA expression between Pim dKO and WT versus average mRNA expression (TPM). mRNA expression (Transcripts per million, TPM) of (**F**) secondary lymphoid homing receptors Sell, Ccr7, S1pr1 and (**G**) key transcription factors involved in CD8 T cell memory differentiation and maintenance Tcf7, Klf2, Foxo1, Foxo3, Id3. (**H**) WT vs Pim dKO protein copy numbers, differentially expression proteins (FC >1.5, q<0.05) are highlighted in red (**I**) Protein copy numbers per cell for key mitochondrial proteins DRP1, OPA1, and CPT1A. (**J**) WT vs Pim dKO protein copy numbers, mitochondrial proteins (as defined in MitoCarta 3.0) are highlight in pink. Symbols in bar charts represent biological replicates, symbols in (**C, E, H, J**) represent the mean. Error bars show mean ± S.D. Data are representative of (**A**) n=3 or show pooled data from (**C**) n=4, and (**D**) n=5 biological replicates with data collected over at least two independent experiments. Quantitative proteomics and RNAseq was performed on biological triplicates. ** q≤0.01, fold-change (FC) shown on bar graphs when q<0.05.

The online version of this article includes the following source data for figure 2:

**Source data 1.** PDF files containing labelled and uncropped images for western blots displayed in *Figure 2A*.

**Source data 2.** Original files for western blot images displayed in *Figure 2A*.

**Source data 3.** Raw values plotted in *Figure 2*.

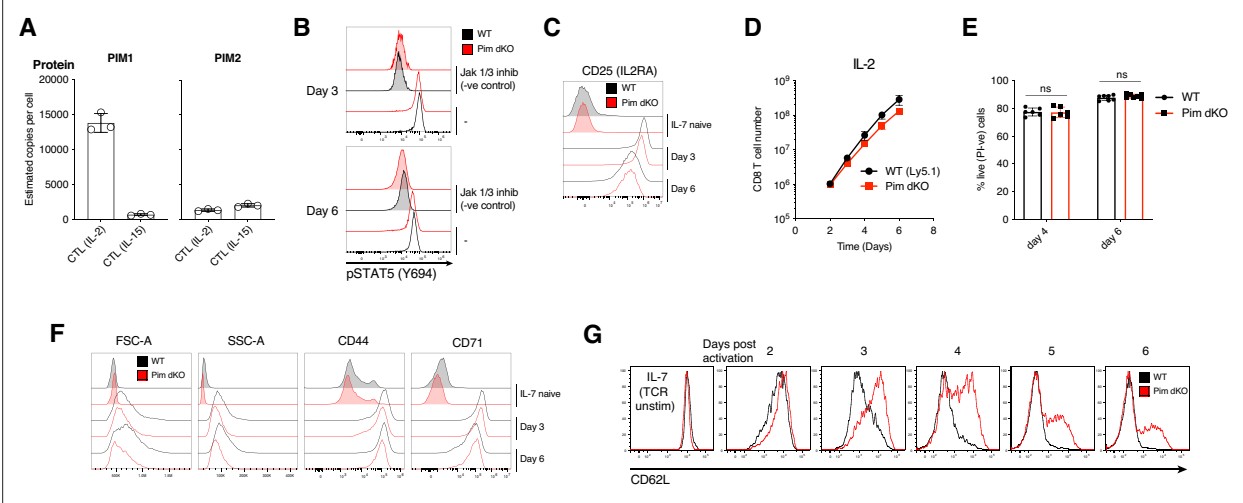

**Figure 3.** Pim dKO IL-2 differentiated effector T cells have reduced cell size and sustained expression of CD62L. (**A**) Estimated copies per cell of PIM1 and PIM2 protein from published quantitative proteomics analysis *Howden et al., 2019*; *Brenes et al., 2023* of CD8 T cells expanded in IL-2 or IL-15 as outlined in *Figure 2B*. (**B–D, F, G**) WT (Ly5.1) and Pim dKO lymph node or spleen single-cell suspensions were mixed at a 50:50 ratio of T cells, activated for 2 days with αCD3/αCD28 (both 0.5 µg/mL) and IL-2 (20 ng/mL), washed then split into fresh medium containing IL-2 (20 ng/mL) daily (as per *Figure 2B*). Some of the mixed cell suspensions were also cultured in IL-7 (5 ng/mL) to sustain a naive T cell reference. (**B**) WT and Pim dKO CTL were treated 1 hr +/- Jak1/3 inhibitor Tofacitinib (100 nM; negative control) before pSTAT5 Y694 expression was measured on day 3 and 6 of culture, (**C**) surface CD25 expression was measured on days 3 and 6 of culture, (**D**) CD8 T cell number vs time was calculated, (**F**) CD8 T cell FSC-A, SSC-A and surface activation markers (CD44, CD71) were measured on days 3 and 6 of culture (**G**) expression of adhesion molecule CD62L was measured daily. (**E**) WT and Pim dKO T cells were activated and expanded with IL-2 as per (**B-D**) and (**F, G**) except in separate cultures and % live cells (PI-ve) was assessed on days 4 and 6 (two-way ANOVA). Symbols in bar charts represent biological replicates, symbols in (**D**) represent the mean. Error bars show mean ± S.D. Data are representative of (**B, G**) n=4, (**C, F**) n=6 or show pooled data from (**D**) n=4, (**E**) n=6 biological replicates with data collected over at least two independent experiments.

The online version of this article includes the following source data for figure 3:

**Source data 1.** Raw values plotted in *Figure 3*.

---

*2020*). These data show that PIM1 and PIM2 do not drive IL-15-mediated metabolic or differentiation programs in antigen-primed CD8 T cells.

## PIM kinases selectively modulate IL-2 controlled effector CD8 T cell differentiation

IL-2 promotes the expression of PIM1 and PIM2 kinases (*Figure 2A*), and in this context, we noted that the copy numbers per cell of PIM1 protein is considerably higher in actively expanding antigen-activated CD8 cells maintained with IL-2 compared to IL-15 (*Figure 3A*). We therefore examined the role of PIM1 and PIM2 in the IL-2 signalling pathways that control the differentiation of effector CTL. The data show that Pim dKO antigen activated CD8 T cells responded normally to IL-2 to induce STAT5 Y694 phosphorylation (*Figure 3B*). They also express normal levels of the IL-2-receptor alpha chain, CD25, (*Figure 3C*), which is sustained by STAT5. Antigen activated WT CD8 T cells have a strong survival and proliferative response to IL-2 producing an ~270-fold expansion between days 2 and 6 of culture in saturating IL-2 concentrations (*Figure 3D*). There was no impact of *Pim1/Pim2*-deficiency on the ability of IL-2 to maintain CD8 T cell viability (*Figure 3E*) and Pim dKO cells CD8 T cells cultured in IL-2 proliferated robustly. They did however exhibit a very mild proliferative disadvantage resulting in an ~2-fold difference in total cell number by day 6 of the culture (*Figure 3D*). We then examined whether *Pim1/Pim2*-deficiency had any impact on the differentiation of Pim dKO CD8 T cells into effector CTL. Expression of activation markers CD44 and CD71 in IL-2-expanded CTL were similar between WT and Pim dKO CD8 T cells on days 3 and 6 of co-cultures (*Figure 3F*). However, one consistent difference between IL-2 maintained WT and Pim dKO CD8 T cells is that WT CTL downregulated expression of CD62L (L-selectin, *Sell*), whereas the Pim dKO CTL sustained higher CD62L expression (*Figure 3G*). We also noted that IL-2 maintained Pim dKO CTL had reduced

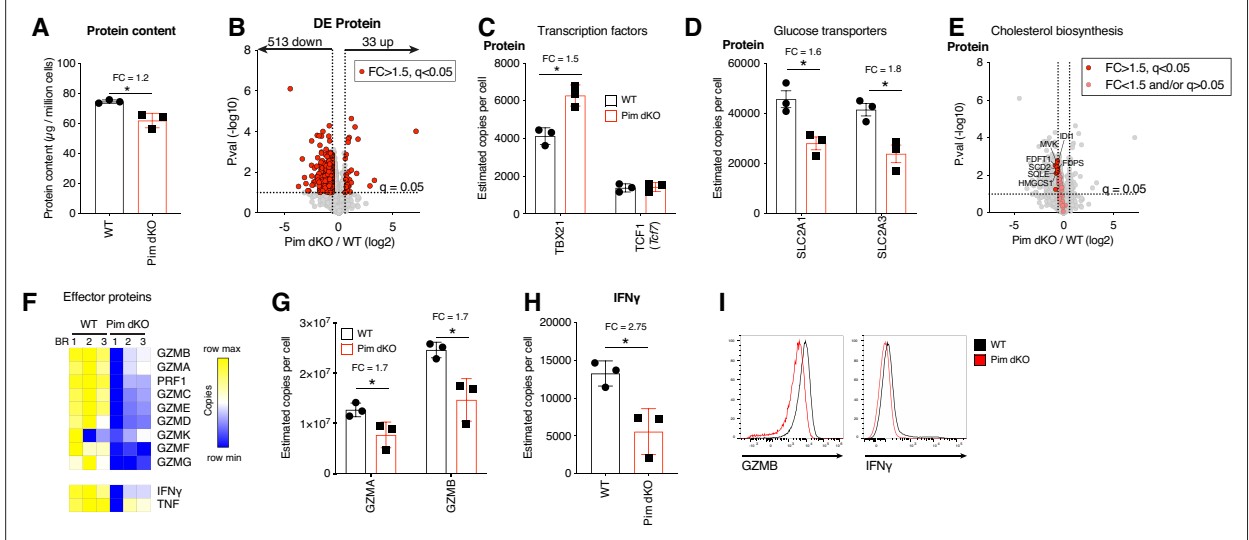

**Figure 4.** Major glucose transporters and effector proteins are reduced in Pim dKO IL-2 expanded CTL. WT and Pim dKO CD8 T cells were activated for 2 days with αCD3/αCD28 (both 0.5 μg/mL) and IL-2 (20 ng/mL), washed then split into fresh medium containing IL-2 (20 ng/mL) daily (as per *Figure 2B*), with an additional CD4 T cell magnetic depletion step on day 3 of culture. CD8 T cells were harvested on day 6 of culture for high-resolution mass spectrometry. An interactive version of the proteomics expression data is available for exploration on the Immunological Proteome Resource website: immpres.co.uk (**A**) Estimated total protein content per cell (student t-test). (**B**) Volcano plots of Pim dKO vs WT protein copy numbers, differentially expressed proteins (FC >1.5, q<0.05) are highlighted in red. Estimated protein copy number per cell of (**C**) transcription factor TBX21 and TCF1 (**D**) glucose transporters SLC2A1 and SLC2A3. (**E**) Volcano plots of Pim dKO vs WT protein copy numbers. Proteins with KEGG term = 'terpenoid backbone biosynthesis', 'biosynthesis of unsaturated fatty acids' or 'steroid biosynthesis' are highlighted with proteins with FC >1.5, q<0.05 shown in red and proteins with FC <1.5 and/or q>0.05 shown in pink. (**F**) Heatmap of protein copy numbers for granzymes, perforin, and effector cytokines. Estimated protein copies for (**G**) major cytolytic Granzymes A and B and (**H**) IFNγ. (**I**) Granzyme B and IFNγ expression was measured by flow cytometry in day 6 IL-2 expanded WT and Pim dKO CTL. Symbols in bar charts show biological replicates. Error bars show mean ± S.D. Data are representative of (**I**) n=3–4, with data collected over at least two independent experiments. Quantitative proteomics was performed on biological triplicates. * indicates q<0.05, fold-change (FC) shown on graph when q<0.05.

The online version of this article includes the following source data and figure supplement(s) for figure 4:

**Source data 1.** Raw values plotted in *Figure 4*.

**Figure supplement 1.** Pim dKO IL-2 expanded CD8 T cells exhibit an effector-like mitochondrial proteome profile.

**Figure supplement 1—source data 1.** Raw values plotted in *Figure 4—figure supplement 1*.

**Figure supplement 2.** 24 hr treatment of IL-2 CTL with pan-PIM kinase inhibitors PIM447 or AZD1208 recapitulates many features of Pim1/Pim2-deficiency.

**Figure supplement 2—source data 1.** Raw values plotted in *Figure 4—figure supplement 2*.

forward and side scatter properties, revealing that they were smaller and less granular than the corresponding WT CTL (*Figure 3F*).

To explore the apparent size change between Pim dKO and WT IL-2 expanded CTL, we used high resolution mass spectrometry to compare the proteomes of day 6 IL-2 cultured WT and Pim dKO CTL. These analyses quantified expression of ~6900 proteins and showed a small decrease in the total protein mass of IL-2 maintained Pim dKO CD8 T cells compared to WT cells (*Figure 4A*, *Supplementary file 4*). The data showed down-regulation of the expression of 513 proteins (>1.5 fold-change, q<0.05) in Pim dKO CTL along with increased copy numbers of 33 proteins (*Figure 4B*, *Supplementary file 4*). The transcription factor profile of Pim dKO T cells was consistent with that of an effector CTL with high levels of TBX21 and low levels of TCF1 (*Tcf7*; *Figure 4C*). Pim dKO T cells also exhibited an effector-like mitochondrial proteome profile (*Figure 4—figure supplement 1*). However, there were salient differences in proteins important for effector functionality between WT and Pim dKO CD8 T cells. These included lower levels of the glucose transporters, SLC2A1 and SLC2A3 (GLUT1 and GLUT3; *Figure 4D*) which are the major glucose transporters in CD8 T cells (*Hukelmann et al., 2016*) and are key for enabling T cells to fuel high rates of glycolysis and protein O-GlcNAcylation required to support effector functions (*Chang et al., 2015*; *Swamy et al., 2016*). Pim dKO CTL also

had decreased expression of a number of proteins involved in fatty acid and cholesterol biosynthesis including SCD2, HMGCS1, MVK, IDI1, FDPS, FDFT1, and SQLE (*Figure 4E*). Another striking observation was that the Pim dKO CD8 cells had reduced levels of a number of key effector proteins including multiple granzymes, perforin, IFNγ, and TNFα (*Figure 4F–H*). Orthogonal flow cytometric analysis of Granzyme B and IFNγ levels confirmed reduced levels of these effector molecule in IL-2 expanded Pim dKO CTL, although IFNγ protein is only lowly produced in IL-2 maintained CTL (*Figure 4I*). In parallel experiments, we also examined the impact of an acute loss of PIM kinase activity on the proteomes of IL-2 maintained CTL to see if this could recapitulate key observations from Pim dKO T cells. When IL-2 maintained WT CTL were treated for 24 hr with the pan-PIM kinases inhibitors PIM447 or AZD1208 we saw reductions in cell proliferation, cell size, cell granularity, fatty acid biosynthesis proteins SCD1-3, glucose transporters SLC2A1, SLC2A3 and a reduction in Granzyme B expression (*Figure 4—figure supplement 2A–E*, *Supplementary file 5*). These experiments corroborated some key observations from the experiments with Pim dKO CTL, where *Pim1* and *Pim2* had been deleted over the lifespan of the T cell.

To examine the molecular regulation underpinning PIM1 and PIM2 kinase control of key metabolic and effector proteins, we also performed RNAseq on IL-2 differentiated WT and Pim dKO CTL samples collected in parallel with the proteomics analysis described in *Figure 4*. In these experiments, ~14,000 polyA+ RNA were quantified and using a cut off >1.5 fold-change, q<0.05 we saw that the abundance of 223 mRNA were decreased and 155 increased by *Pim1/Pim2* deficiency (*Figure 5A*, *Supplementary file 6*). These data revealed that the reduced expression of perforin and Granzymes C-G and K protein corresponded to decreases in their mRNA (*Figure 5B and C*). However, there were striking examples where a clear decrease in protein levels did not correspond with an appreciable decrease in mRNA expression. These included the predominant cytolytic Granzymes B and A (*Figure 5C and D*), as well as both glucose transporters SLC2A1 and SLC2A3 (*Figure 5E*). Furthermore, contrasting differential expression of proteins and mRNA data in Pim dKO vs WT CTL revealed that ~75% of the proteins whose expression was changed in Pim dKO T cells (FC >1.5, q<0.05), exhibited no strong corresponding changes in their mRNA (FC <1.2; *Figure 5F*, *Supplementary file 7*). Indeed, of the ~500 proteins whose expression decreased at the protein level in Pim dKO CTL only 17 showed a strong decrease in magnitude at the mRNA level as well (*Figure 5G*, *Supplementary file 7*).

These observations could reflect that the PIM kinases are required for IL-2 to maximally control protein synthesis in CD8 T cells. To examine this hypothesis, we first interrogated our proteomics and transcriptomics data. Consistent with this hypothesis, we observed a small reduction in ribosome protein content in Pim dKO CTL, which scaled with the reduction in total protein content (*Figure 5H*). Moreover, the translational repressor PDCD4 was increased in Pim dKO CTL (*Figure 5I*). This could be confirmed by flow cytometry (*Figure 5J*) and was underpinned at the mRNA level (*Figure 5B*). Similarly, 24 hr inhibition with pan PIM kinases inhibitors was sufficient to substantially increase PDCD4 protein expression (*Figure 4—figure supplement 2F*). We also noted that there was a small decrease in the expression of EIF4A1, a key component of the eIF4F translation initiation complex, in Pim dKO IL-2 maintained CTL (*Figure 5K*). PDCD4 inhibits translation by binding EIF4A1 in a 2:1 ratio, preventing its interaction with the EIF4F complex (*Suzuki et al., 2008*). Together the changes in both proteins equated to an approximate halving of the ratio between EIF4A1 and PDCD4 (*Figure 5L*). These data all support that loss of PIM kinases could reduce protein synthesis in IL-2 maintained CTL. To test this hypothesis, we used a single-cell assay that quantifies the incorporation of an analogue of puromycin (OPP) into newly synthesized protein chains in the ribosome to assess cellular rates of protein synthesis. We found that 24 hr of treatment with pan-PIM kinase inhibitors PIM447 or AZD1208 caused a quantitative reduction in protein synthesis rates of IL-2 expanded CTL (*Figure 5M*). This data is consistent with PIM kinase effects on protein expression being via control of protein translation.

## PIM1 and PIM2 regulate mTORc1 and sustain lymphoid homing in Pim1/Pim2-deficient effector CTL

The pattern of proteome remodelling in Pim dKO CTL had some phenotypic features in common with the previously reported changes that occur following inhibition of the Serine/Threonine kinase mTORc1 in CTL (*Hukelmann et al., 2016*; *Howden et al., 2019*). These included: increased expression of PDCD4 and decreased expression of cholesterol biosynthesis enzymes and effector molecules including granzymes, perforin, IFNγ and TNFα (*Hukelmann et al., 2016*; *Howden et al., 2019*). In

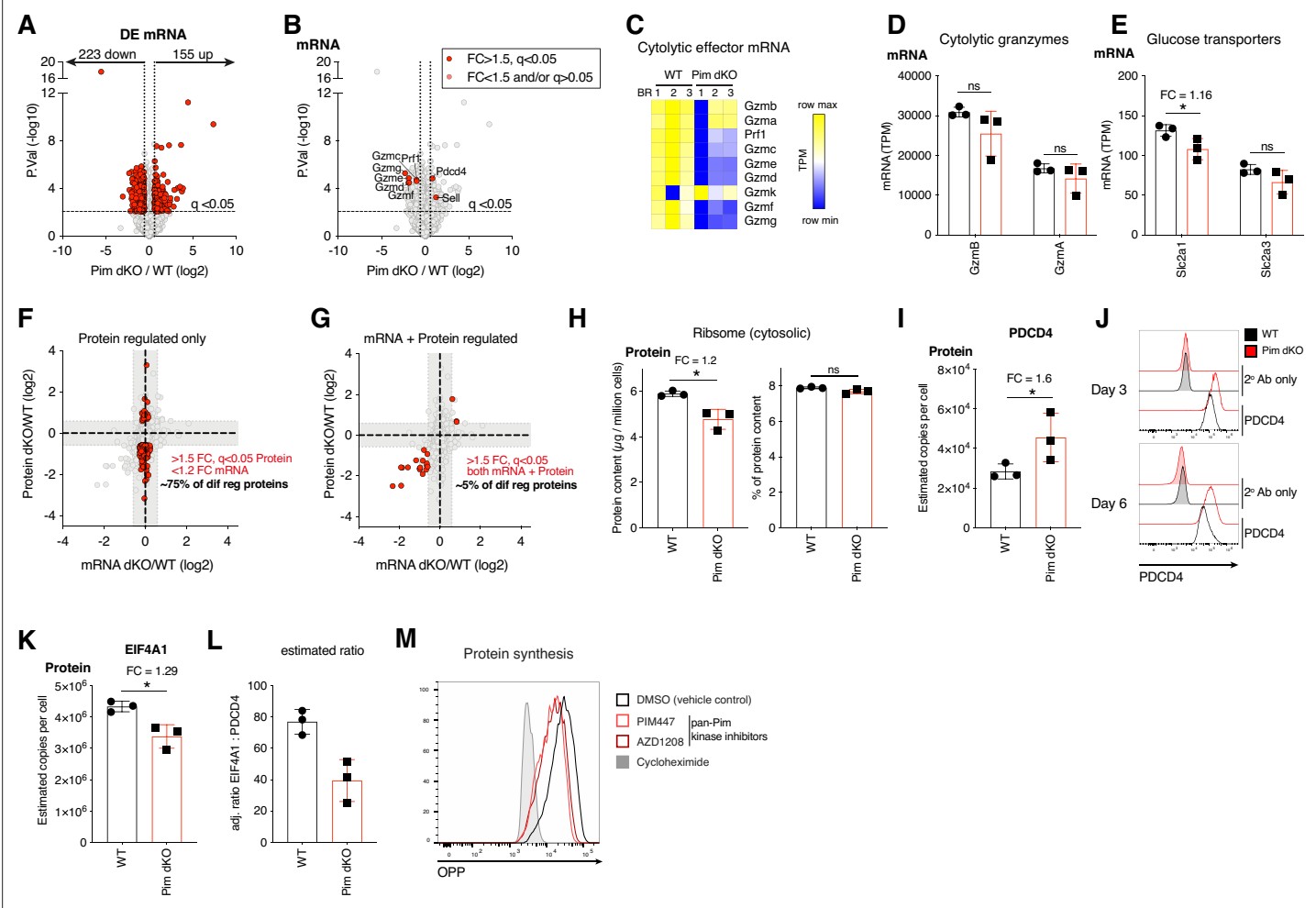

**Figure 5.** Disconnect between protein and mRNA expression in Pim1/Pim2-deficient effector CTL corresponds with a reduction in protein synthesis. RNAseq analysis was performed in day 6 IL-2 expanded WT and Pim dKO CD8 T cells which were collected in parallel with proteomics analysis described in *Figure 4A*. (**A**) Volcano plot of RNAseq data, differentially expressed mRNA (FC >1.5, q<0.05) are highlighted in red. (**B**) Volcano plot of RNAseq data, Granzymes C-K, perforin, Pdcd4 and Sell are highlighted in red. (**C**) Heatmap of mRNA expression (TPM) for granzymes, perforin and effector cytokines. Bar chart of mRNA expression (TPM) of (**D**) Granzymes A and B (**E**) Glucose transporters Slc2a1 and Slc2a3. (**F, G**) Fold change of PimdKO/WT protein from proteomics analysis described in *Figure 4* vs mRNA (**F**) highlighting in red proteins that are differentially expressed (FC >1.5, q<0.05) where mRNA is not substantially different (FC <1.2) and (**G**) highlighting in red protein and mRNA that are both differentially expressed (FC >1.5, q<0.05). (**H**) Estimated cytosolic ribosome content per cell (left), % ribosome of total cellular protein content (right). (**I**) Estimated protein copy number per cell of translation repressor PDCD4. (**J**) PDCD4 expression measured by flow cytometry on day 3 and 6 in IL-2 expanded WT vs Pim dKO CD8 T cells. (**K**) Estimated protein copy number per cell of EIF4A1. (**L**) Adjusted ratio of PDCD4: EIF4A1 (assuming 1 PDCD4 binds 2 x EIF4A1) in WT and Pim dKO proteomes. (**M**) Protein synthesis measured by OPP incorporation in day 6 IL-2-expanded WT CTL treated for 24 hours with pan PIM kinase inhibitors PIM447 (5 μM) or AZD1208 (10 μM). 30-min cycloheximide (100 μg/mL) treatment gives no protein synthesis background control. Symbols in bar charts show biological replicates: error bars show mean ± S.D. Data are representative of (**M**) n=2 biological replicates collected over two independent experiments, (**J**) n=2 biological replicates. Quantitative proteomics and RNAseq were performed on biological triplicates. * indicates q<0.05, fold-change (FC) shown on graph when q<0.05.

The online version of this article includes the following source data and figure supplement(s) for figure 5:

**Source data 1.** Raw values plotted in *Figure 5*.

**Figure supplement 1.** Pim dKO IL-2 expanded CD8 T cells are more similar to IL-2 WT effector T cells than IL-15 expanded memory T cells.

this respect, mTORc1 activity in CTL is acutely sensitive to reduced availability of glucose (*Rolf et al., 2013*) and the reduced expression of glucose transporters in Pim dKO T cells might predict that these cells would have lower activity of mTORc1. We therefore assessed if PIM kinases were required for IL-2 activation of mTORc1. In these experiments, we compared the phosphorylation of S6K1 (p70 S6-kinase 1) on its mTORC1 substrate sequence T389 in WT and Pim dKO CD8 T cells by western

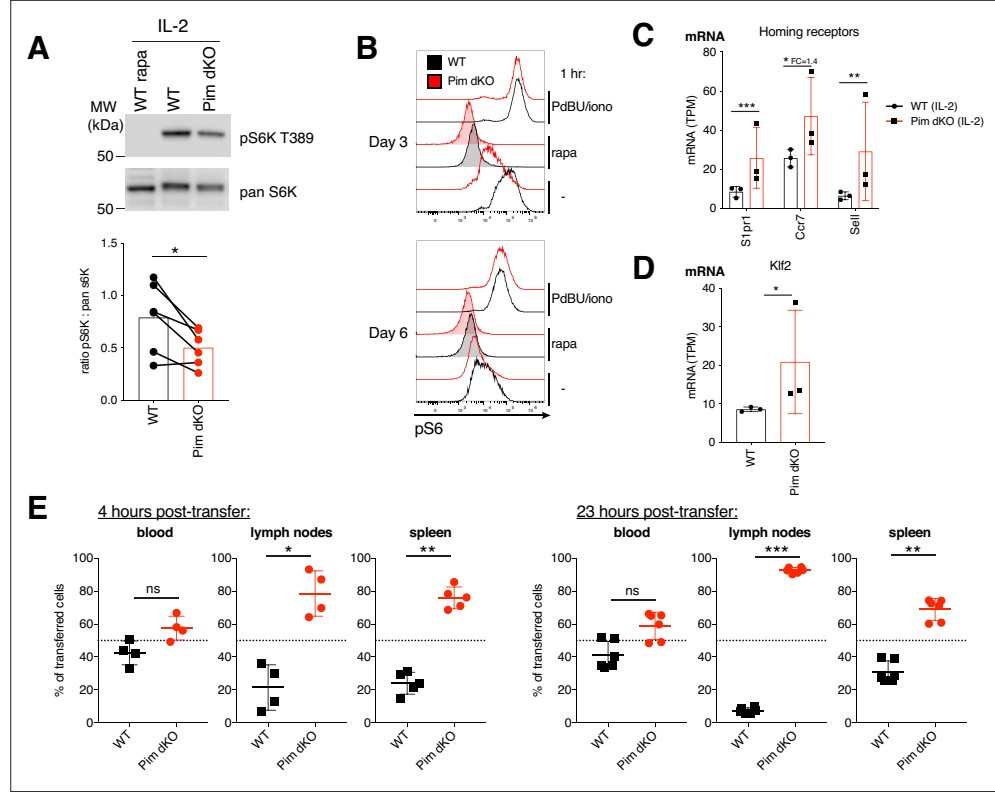

**Figure 6.** PIM kinases regulate mTORc1 activity and lymphoid homing in effector CTL. (**A**) Western blot of pS6K T389 and pan S6K from day 6 WT and PimdKO IL-2 CTL (paired student t-test, *=p < 0.05). (**B**) WT and Pim dKO T cells were mixed in a 50:50 ratio, activated with αCD3/αCD28 (both 0.5 µg/mL) and IL-2 (20 ng/mL) and expanded in IL-2 as per *Figure 2B* and pS6 (Ser235/236) measured after 1 hr +/- PdBU and ionomycin (positive control), +/-rapamycin (mTORc1 inhibitor, negative control) or no additional treatment. mRNA expression (TPM) from RNAseq analysis of IL-2 expanded WT and Pim dKO CTL described in *Figure 5A* for (**C**) cell homing receptors S1pr1, Ccr7 and Sell and (**D**) transcription factor Klf2. (**E**) WT and Pim dKO T cells were activated and expanded with IL-2 in separate cultures as per *Figure 2B*. On day 6 of culture WT and Pim dKO CTL were labelled with CFSE or CTV, mixed at a 50:50 ratio and transferred into C57BL/6-recipient mice. Values indicate percentage of transferred cells detected in blood, lymph node or spleen 4 or 23 hr post-transfer that were WT or Pim dKO (one-way ANOVA). Symbols show biological replicates. Error bars show mean ± S.D. Data are representative of (**A**) n=6, (**B**) n=2–4 collected across at least two independent experiments and (**E**) n=6 recipient mice, from n=2 biological donor replicates. RNAseq was performed on biological triplicates. * q<0.05, ** q<0.01, *** q<0.001, fold-change (FC) shown on graph when q<0.05 but FC <1.5.

The online version of this article includes the following source data for figure 6:

**Source data 1.** PDF files containing labelled and uncropped images for western blots displayed in *Figure 6A*.

**Source data 2.** Original files for western blot images displayed in *Figure 6A*.

**Source data 3.** Raw values plotted in *Figure 6*.

blot. These data show a modest reduction of mTORc1 activity in the Pim dKO T cells (*Figure 6A*). We also assessed the activity of S6K1 at a single-cell level by quantifying the phosphorylation of its downstream target protein S6 ribosomal protein on S235/S236 by flow cytometry over the course of IL-2 driven CTL expansion. These data also show that *Pim1/Pim2*-deficiency reduced IL-2 induced mTORc1-dependent phosphorylation of S6 but did not compromise the ability of the Pim dKO cells to phosphorylate S6 in response to phorbol esters and calcium ionophores; pharmacological activators that induce RSK mediated S6 phosphorylation (*Figure 6B*).

Previously, it has been shown that decreased glucose availability or inhibition of mTORc1 prevents the downregulation of CD62L that normally accompanies IL-2-induced CTL differentiation (*Sinclair et al., 2008*; *Finlay et al., 2012*). Thus, the reduced expression of glucose transporters and reduced activity of mTORc1 in Pim dKO IL-2 expanded CTL provides a potential explanation for the striking phenotype shown in

*Figure 3G*, that IL-2 maintained Pim dKO CTL retain high levels of CD62L/L-selectin. The IL-2 CTL transcriptomics data showed there is increased expression of CD62L at the mRNA level in IL-2 maintained PimdKO CTL (*Figures 5B and 6C*). The expression of *Sell* (CD62L) mRNA is controlled by the transcription factor KLF2 and the expression of *Klf2* mRNA is also increased in IL-2 maintained Pim dKO CTL (*Figure 6D*) as are mRNA levels for other KLF2 targets: the chemokine receptors *Ccr7* and *S1pr1* (*Figure 6C*).

CD62L/L-selectin controls T cell adhesion to the endothelium of high endothelial venules and is essential for lymphocyte transit from blood into secondary lymphoid tissue such as lymph nodes. CCR7, and the S1P1 receptor also direct migration by controlling T cell entry and egress respectively into secondary lymphoid tissue. In this context, the loss of CD62L, CCR7 and S1PR1 expression following immune activation reprograms the trafficking of effector T cells away from lymphoid tissue toward sites of inflammation in peripheral tissues (*Sinclair et al., 2008*; *Nolz et al., 2011*). Accordingly, the differences between Pim dKO and WT CTL in terms of CD62L expression could indicate that in addition to having reduced effector protein expression, Pim dKO CTL also do not switch their trafficking behaviour to that of a fully differentiated effector CTL and rather retain the capacity to home to lymphoid tissues. To test this possibility in vivo, we performed adoptive transfer experiments comparing the ability of WT effector CTL and Pim dKO CTL to home to secondary lymphoid tissues. In these experiments, αCD3/αCD28 activated WT or Pim dKO T cells were differentiated in IL-2 and cells were then labelled with either CFSE or CTV dyes; mixed at a ratio of 1:1, and adoptively transferred into C57BL/6 hosts. *Figure 6E* shows that Pim dKO CTL, but not WT CTL, retained substantial capacity to home to secondary lymphoid organs and, hence, accumulated in lymph nodes and spleen: PIM kinases are thus required for the normal reprogramming of CTL trafficking.

## Discussion

The objective of the present study was to use quantitative analysis of T cell proteomes and transcriptomes to explore how deletion of the Serine/Threonine kinases PIM1 and PIM2 impacts CD8 T cell activation and differentiation. The expression of PIM1 and PIM2 is induced during immune activation of CD8 T cells by antigen and co-stimulation and then sustained to varying degrees by the cytokines IL-2 or IL-15. The loss of PIM1 and PIM2 was shown to have no effect on the CD8 T cell programs controlled by antigen receptor/co-stimulation and had very little impact on the IL-15-driven transcriptional and proteomic programs in memory differentiated CD8 T cells. However, PIM1 and PIM2 were required for IL-2 programming of effector CD8 T cell differentiation. IL-2 controls essential cytolytic effector and metabolic programs in CTL, is a key cytokine for CD8 T cell anti-viral immunity and is also used to produce cytotoxic T cells for anti-cancer immunotherapy (*Kalia and Sarkar, 2018*). The understanding that PIM kinases mediate IL-2 control of the expression of glucose transporters, lipid metabolic enzymes and key cytolytic proteins such as granzymes A and B in effector CD8 T cells is thus fundamental information about how this key cytokine controls T cells. It was also important to understand that PIM kinases did not control the transcriptional programs that direct expression of glucose transporters and effector enzymes and hence the full consequences of PIM kinase deletion could only be assessed by in depth analysis of the proteomes of *Pim1/Pim2*-deficient CTL and not from mRNA sequencing. The loss of PIM kinase activity reduced protein synthesis rates in CTL which explains the discordance between mRNA and protein reported herein. The reduction in protein synthesis was modest but previous studies have shown that modest reductions in protein synthesis and loss of mTORc1 activity impact CD8 T cell differentiation programs (*Cornish et al., 2006*; *Hukelmann et al., 2016*; *Howden et al., 2019*). PIM kinase regulation of protein translation has been reported previously in mouse B cell lymphoma and mouse embryonic fibroblasts (*Schatz et al., 2011*) and some of the key examples of proteins we found to be unaltered at the mRNA level but down-regulated at the protein level have prior reports of sensitivity to translational regulation in immune cells (*Fehniger et al., 2007*; *Ricciardi et al., 2018*). Reduced protein synthesis capacity in cells lacking PIM kinases could be caused by reduced mTORc1 activity combined with increased expression at the mRNA and protein level of the translational repressor PDCD4. Moreover, protein synthesis is highly energetically demanding and once CTL have reduced glucose transporter expression this would reduce glucose availability to fuel protein production. These data collectively point towards modulation of protein translation as an important target of PIM kinase activity downstream of IL-2 in CD8 T cells.

One other important insight from the current work is that PIM kinases are required for IL-2 to fully repress expression of the adhesion molecule CD62L and the chemokine receptors CCR7 and S1PR1.

These molecules play a key role in controlling T cell homing to secondary lymphoid tissue and are expressed at high levels in naive and memory T cells. IL-2 induces down regulation of CD62L, CCR7, and S1PR1 expression as part of a program that causes effector cells to lose their capacity to home to secondary lymphoid tissues (*Sinclair et al., 2008*; *Nolz et al., 2011*). The present data show that PIM kinases can mediate this IL-2 repression of lymph node homing receptors and the ability of IL-2 to switch off the naive T cell trafficking program. IL-2 maintained Pim dKO CD8 effector T cells thus retain the capacity to home to secondary lymphoid tissues in vivo. This lymph-node homing pattern is also a memory CD8 T cell phenotype but here it is pertinent that the current in-depth analysis of the transcriptomes and proteomes of *Pim1*/*Pim2*-deficient IL-2 differentiated CD8 T cells show that the loss of PIM kinases does not cause effector T cells to differentiate instead into memory cells. For example, IL-2 maintained Pim dKO CD8 T cells do not re-express key molecules that maintain the stem-like properties of memory cells such as the transcription factor TCF7 and IL-7 receptor (*Supplementary file 2* and *Supplementary file 6*). Nor do Pim dKO CTL cultured in IL-2 have the mitochondrial proteome of a memory T cell and PCA plots of IL-15 and IL-2 proteomics and RNAseq data show that Pim dKO IL-2 expanded CTL are still much more similar to IL-2 expanded WT CTL than to IL-15 expanded CTL (*Figure 5—figure supplement 1*). Hence *Pim1*/*Pim2*-deficiency prevents optimal differentiation of effector CTL but does not cause these cells to fully switch to a memory phenotype. This is worth knowing as pan-PIM kinase inhibitors are in development as anti-cancer drugs and it has been suggested that PIM inhibitors might be used as a strategy to generate more stem-memory-like CD8 T cells in combination with cancer immunotherapy treatments (*Chatterjee et al., 2019*; *Clements and Warfel, 2022*). Information about how the loss of PIM kinases will impact CD8 T cells is very relevant to informed usage of such drugs.

Finally, the selectivity of PIM kinases for IL-2 control of conventional CD8 T cell differentiation with no evidence that PIMs were important for IL-15 responses beyond a modest effect on cell proliferation was interesting. This failure to see a role for PIM signalling in conventional peripheral CD8 T cell differentiation is in contrast to observations that PIM1 and PIM2 direct IL-15 control of the metabolic programming of intestinal IEL (*James et al., 2021*). A notable difference between these scenarios was that in IEL, IL-15 was being used individually to drive immune activation and was sufficient to strongly induce PIM1 and PIM2 expression. In contrast, in our conventional CD8 T cell system we uncovered that IL-15 does not sustain high levels of PIM kinases, which could possibly explain this discrepancy. However, high levels of PIM expression alone are not inevitably predictive of their importance. In this study, we observed very high levels of both PIM1 and PIM2 in antigen activated CD8 T cells yet loss of PIM kinases there had no functional impact. Previously, it was suggested that mTORc1 could compensate for PIM kinase deficiency (*Fox et al., 2005*). However, we found no evidence for this, and in fact observed that PIM kinases were required to maximally stimulate mTORc1 activity in IL-2 expanded CTL. Thus, this examination of the role of PIM kinases in multiple signalling situations reveal cell context-dependent regulatory roles for PIM kinases rather than hard wired functions that apply to every stimulatory situation in T cells.

## Materials and methods

**Key resources table**

| Reagent type (species) or resource | Designation | Source or reference | Identifiers | Additional information |
|---|---|---|---|---|
| Genetic reagent (*Mus musculus*) | *Pim1KO/Pim2KO* (*Pim1/2* dKO) | PMID:8233823, PMID:15199164 | | *Pim1* and *Pim2* KO strains generated on the FVB/N background in references listed and backcrossed to C57BL/6 background in this paper |
| Genetic reagent (*M. musculus*) | C57BL/6 J (WT) | Charles River UK | | |
| Genetic reagent (*M. musculus*) | TCR α-; P14 TCRVα2Vβ8 (P14) | PMID:2573841 | | maintained in house as an P14 TCR transgene heterozygote |

*Continued on next page*

*Continued*

| Reagent type (species) or resource | Designation | Source or reference | Identifiers | Additional information |
|---|---|---|---|---|
| Genetic reagent (*M. musculus*) | C57BL/6-Tg(TcraTcrb)1100Mjb (OT1) | PMID:8287475 | | maintained in house as an OT1 TCR transgene heterozygote on a CD45.1 (Ly5.1) background |
| Genetic reagent (*M. musculus*) | C57BL/6 J Ly5.1 (Ly5.1) | Charles River UK | | |
| Antibody | Anti-CD3 (armenian hamster, monoclonal, 145–2C11) | Thermo Fisher Scientific | Cat # 14-0031-82, RRID:AB_467049 | T cell culture: 0.5 µg/mL |
| Antibody | Anti-CD28 (syrian hamster, monoclonal, 37.51) | Thermo Fisher Scientific | Cat # 16-0281-82, RRID:AB_468921 | T cell culture: 0.5 µg/mL |
| Antibody | Anti-CD4 biotin (Rat, monoclonal, RM4-5) | Biolegend | Cat # 100508, RRID:AB_312710 | CD4 T cell depletion: 5 µg/mL |
| Antibody | Anti-mouse CD16/CD32 Fc Block, (rat, monoclonal) | BD Biosciences | Cat # 553141, RRID:AB_394656 | Fc block: 1:60 |
| Antibody | anti-mouse CD4 (Rat, monoclonal, RM4-5) | Thermo Fisher Scientific/ eBioscience | Cat # 47-0042-82, RRID:AB_1272183 | cell surface stain 1:200, APC eF780 |
| Antibody | anti-mouse CD4 (Rat, monoclonal, RM4-5) | BD Biosciences | Cat # 560782, RRID:AB_1937327; Cat # 553650, RRID:AB_394970; Cat # 552775, RRID:AB_394461 | cell surface stain 1:200, V500, FITC, PECy7 |
| Antibody | anti-mouse CD4 (Rat, monoclonal, RM4-5) | Biolegend | Cat # 100553, RRID:AB_2561388 | cell surface stain 1:200, BV510 |
| Antibody | anti-mouse CD8a (Rat, monoclonal, 53–6.7) | Biolegend | Cat # 100738, RRID:AB_11204079; Cat # 100708, RRID:AB_312747; Cat # 100722, RRID:AB_312761; Cat # 100712, RRID:AB_312750 | cell surface stain 1:200, BV421, PE, PECy7, APC |
| Antibody | anti-mouse CD8a (Rat, monoclonal, 53–6.7) | BD Biosciences | Cat # 551162, RRID:AB_394081 | cell surface stain 1:200, PerCPCy5.5 |
| Antibody | anti-mouse CD8a (Rat, monoclonal, 53–6.7) | Thermo Fisher Scientific/ eBioscience | Cat # 47-0081-82, RRID:AB_1272185 | cell surface stain 1:200, APC eF780 |
| Antibody | anti-mouse CD25 (Rat, monoclonal, 7D4) | BD Biosciences | Cat # 553072, RRID:AB_394604 | cell surface stain 1:200, FITC |
| Antibody | anti-mouse CD25 (Rat, monoclonal, PC61) | Biolegend | Cat # 102016, RRID:AB_312864 | cell surface stain 1:200, PECy7 |
| Antibody | anti-mouse CD44 (Rat, monoclonal, IM7) | Biolegend | Cat # 103044, RRID:AB_2561391; Cat # 103006, RRID:AB_312956; Cat # 103030, RRID:AB_830787 | cell surface stain 1:200, BV510, FITC, PECy7 |
| Antibody | anti-mouse CD44 (Rat, monoclonal, IM7) | Thermo Fisher Scientific/ eBioscience | Cat # 47-0441-82, RRID:AB_1272244 | cell surface stain 1:200, APC eF780 |
| Antibody | anti-mouse CD45.1 (Mouse, monoclonal, A20) | Biolegend | Cat # 110728, RRID:AB_893346; Cat # 110714, RRID:AB_313503 | Cell surface stain 1:200, PerCPCy5.5, APC |
| Antibody | anti-mouse CD62L (Rat, monoclonal, MEL-14) | Thermofisher Scientific/ eBioscience | Cat # 12-0621-83, RRID:AB_465722 | Cell surface stain 1:200, PE |
| Antibody | anti-mouse CD62L (Rat, monoclonal, MEL-14) | Biolegend | Cat # 104435, RRID:AB_10900082; Cat # 104412, RRID:AB_313099 | Cell surface stain 1:200, BV421, APC |
| Antibody | anti-mouse CD69 (Armernian Hamster, monoclonal, H1.2F) | Thermo Fisher Scientific/ eBioscience | Cat # 17-0691-82, RRID:AB_1210795 | Cell surface stain 1:200, APC |
| Antibody | anti-mouse CD71 (Rat, monoclonal, RI7217) | Biolegend | Cat # 113813, RRID:AB_10899739; Cat # 113820, RRID:AB_2728134 | cell surface stain 1:200, BV421, APC |

*Continued on next page*

*Continued*

| Reagent type (species) or resource | Designation | Source or reference | Identifiers | Additional information |
|---|---|---|---|---|
| Antibody | anti-mouse TCRbeta (Armenian Hamster, monoclonal, H57-597) | Thermo Fisher Scientific/ eBioscience | Cat # 45-5961-82, RRID:AB_925763 | cell surface stain 1:200, PerCPCy5.5 |
| Antibody | anti-mouse IFNγ (Rat, monoclonal, XMG1.2) | Biolegend | Cat # 505810, RRID:AB_315404 | intracellular stain 1:100, APC |
| Antibody | anti-mouse Granzyme B (Rat, monoclonal, NGZB) | Thermo Fisher Scientific/ eBioscience | Cat # 12-8898-82, RRID:AB_10870787; Cat # 17-8898-82, RRID:AB_2688068 | intracellular stain 1:200, PE, APC |
| Antibody | anti-phospho S6 ribosomal protein (Ser235/236) Alexa Fluor 647 (Rabbit, monoclonal, D57.2.2E) | Cell Signaling Technology | Cat # 4851 S, RRID:AB_10695457 | intracellular stain 1:100, AF647 |
| Antibody | anti-phospho STAT5 Y694 (Rabbit, monoclonal, C11C5) | Cell Signaling Technology | Cat # 9359 S, RRID:AB_823649 | intracellular stain 1:200; western blot 1:1000 |
| Antibody | anti-PDCD4 (Rabbit, monoclonal, D29C6) | Cell Signaling Technology | Cat # 9535 S, RRID:AB_2162318 | intracellular stain 1:100 |
| Antibody | anti-rabbit IgG Fab2 Alexa Fluor 647 (goat) | Cell Signaling Technology | Cat # 4414 S, RRID:AB_10693544 | intracellular stain 1:1000 |
| Antibody | p70 S6 Kinase Antibody (Rabbit, polyclonal) | Cell Signaling Technology | Cat # 9202 S, RRID:AB_331676 | western blot 1:1000 |
| Antibody | Phospho-p70 S6 Kinase (Thr389) (Rabbit, monoclonal, 108D2) | Cell Signaling Technology | Cat # 9234 S, RRID:AB_2269803 | western blot 1:1000 |
| Antibody | Pim1 antibody (mouse, monoclonal, 12H8) | Santa Cruz | Cat # sc-13513; RRID:AB_628129 | western blot 1:200 |
| Antibody | Pim2 antibody (mouse, monoclonal, 1D12) | Santa Cruz | Cat # SC-13514; SC-13514 | western blot 1:200 |
| Antibody | Anti-rabbit IgG HRP (goat, polyclonal) | Thermo Fisher Scientific | Cat # 31460; RRID:AB_228341 | western blot 1:5000 |
| Antibody | Anti-mouse IgG HRP (horse, polyclonal) | Cell Signaling Technology | Cat # 7076 S; RRID:AB_330924 | western blot 1:5000 |
| Peptide, recombinant protein | recombinant human IL-2 | Novartis, UK | Proleukin | T cell culture: 20 ng/mL |
| Peptide, recombinant protein | recombinant human IL-15 | Peprotech | Cat # 200–15 | T cell culture: 20 ng/mL |
| Peptide, recombinant protein | recombinant mouse IL-7 | Peprotech | Cat # 217–17 | T cell culture: 5 ng/mL |
| Peptide, recombinant protein | recombinant mouse IL-12 | R&D Systems, UK | Cat # 419 ML | T cell culture: 2 ng/mL |
| Peptide, recombinant protein | gp33-41 peptide | Anaspec | Cat #AS-61296 | T cell culture: 100 ng/mL |
| Commercial assay or kit | RNAeasy mini kit | Qiagen | Cat # 74104 | RNA purification |
| Commercial assay or kit | TruSeq Stranded mRNA sample preparation kit | Illumina | Cat # 15031047 | RNASeq library preparation |

*Continued on next page*

*Continued*

| Reagent type (species) or resource | Designation | Source or reference | Identifiers | Additional information |
|---|---|---|---|---|
| Commercial assay or kit | EZQ protein quantitation kit | Thermo Fisher Scientific | Cat # R33200 | Protein quantification |
| Commercial assay or kit | CBQCA protein quantitation kit | Thermo Fisher Scientific | Cat # C6667 | Peptide quantification |
| Commercial assay or kit | eBioscience Intracellular Fixation and Permeabilization Buffer Set | Thermo Fisher Scientific/ eBioscience | Cat # 88-8824-00 | Intracellular stain |
| Chemical compound, drug | PIM447 | MedChemExpress | Cat # HY-19322B | 1 or 5 µM as indicated |
| Chemical compound, drug | AZD1208 | MedChemExpress | Cat # HY-15604 | 1 or 10 µM as indicated |
| Chemical compound, drug | Tofacitinib | Selleckchem | Cat # CP-690550 | 100 nM |
| Chemical compound, drug | Rapamycin | Merck/Calbiochem | Cat # 553211 | 20 nM |
| Chemical compound, drug | PDBu | Cell Signaling Technology | Cat # 12808 | 20 ng/mL |
| Chemical compound, drug | ionomycin | Merck/Calbiochem | Cat # 407951 | 500 ng/mL |
| Chemical compound, drug | DAPI | Thermo Fisher Scientific | Cat # D1306 | 1 µg/mL |
| Chemical compound, drug | Propidium iodide | Sigma | Cat # P4170 | 0.2 µg/mL |
| Chemical compound, drug | GolgiPlug | BD Biosciences | Cat # 555029 | 1:1000 |
| Chemical compound, drug | CellTrace Violet | Invitrogen | Cat # C34557 | 5 µM |
| Chemical compound, drug | CFSE | Invitrogen | Cat # C34554 | 5 µM |
| Chemical compound, drug | O-propargyl puromycin | Jena Bioscience | Cat # NU-931 | 20 µM, 10 min |
| Chemical compound, drug | cycloheximide | Sigma | Cat # C7698 | 100 µg/mL, 30 min |
| Chemical compound, drug | Alexa-647-azide | Thermo Fisher Scientific | Cat # A10277 | 5 µM in click reaction buffer |
| Software, algorithm | FlowJo software | BD Biosciences, developed by Treestar | RRID:SCR_008520 | versions 9.9.6 or version 10.6.1 and above |
| Software, algorithm | Maxquant | https://www.maxquant.org, PMID:19029910 | RRID:SCR_014485 | version 1.6.10.43 |
| Software, algorithm | Perseus | https://www.maxquant.org/ perseus, PMID:27348712 | RRID:SCR_015753 | version 1.6.6.0 |
| Software, algorithm | Spectronaut | Biognosys | | version 14.7 |
| Software, algorithm | Morpheus | https://software.broadinstitute. org/morpheus | RRID:SCR_017386 | heatmap generation |
| Software, algorithm | Prism | GraphPad | RRID:SCR_002798 | version 9 or 10 |
| Software, algorithm | Rstudio | Rstudio | RRID:SCR_000432 | version 1.6.10.43 |

*Continued on next page*

*Continued*

| Reagent type (species) or resource | Designation | Source or reference | Identifiers | Additional information |
|---|---|---|---|---|
| Other | Sera-Mag SpeedBead Carboxylate-modified magnetic particles (hydrophilic) | GE Lifesciences | Cat # 45152105050250 | proteomic sample prep |
| Other | Sera-Mag SpeedBead Carboxylate-modified magnetic particles (hydrophobic) | GE Lifesciences | Cat # 65152105050250 | proteomic sample prep |
| Other | S-Trap Mini columns | Protifi | Cat # CO2-mini-80 | proteomic sample prep |
| Other | RapidSphere Beads | EasySep | Cat # 50001 | CD4 T cell depletion |
| Other | sphero rainbow beads | BD Biosciences | Cat # 556288 | cell counting, 10,000 beads/sample |

## Mice

Pim1$^{-/-}$ (Pim1 KO) mice (*Laird et al., 1993*) and Pim2$^{-/-}$ or Pim2$^{-/Y}$ (Pim2 KO) mice (*Mikkers et al., 2004*) on the FVB/N background were backcrossed for >10 generations onto a C57BL/6 background and were a generous gift from Prof Victor Tybulewicz. Pim1 KO and Pim2 KO mice were inter-crossed to generate the Pim1/Pim2 double KO (Pim dKO) strain. Pim dKO, C57BL/6, Ly5.1, P14 and OT1 mice were bred and maintained in the WTB/RUTG, University of Dundee. Mice used for proteomics and RNAseq studies were male and 12–15 weeks of age. Age/sex matched mice were used for all other experiments between 5 and 52 weeks of age, with most being between 10 and 20 weeks. All animal experiments were performed under Project License PPL 60/4488 and P4BD0CE74.The University of Dundee Welfare and Ethical Use of Animals Committee accepted the project licence for submission to the UK Home Office. All studies, breeding and maintenance performed in Dundee in compliance with UK Home Office Animals (Scientific Procedures) Act 1986 guidelines. Individual study plans were approved and deemed compliant by the UVS/Named Compliance Officer.

## Cell culture

Single-cell suspensions were generated by mashing mouse lymph nodes (brachial, axial, inguinal, superficial cervical, deep cervical, lumbar) or spleens through a 70 µm strainer. Red blood cells were lysed with Ack buffer (150 mM NH$_4$Cl 10 mM KHCO$_3$ 110 µM Na$_2$EDTA pH 7.8). Cells were cultured in RPMI 1640 containing glutamine (Invitrogen), supplemented with 10% FBS (Gibco), penicillin/streptomycin (Gibco) and 50 µM β-mercaptoethanol (Sigma) at 37 °C with 5% CO$_2$.

For 24 hr TCR activation WT and Pim dKO proteomics and phenotyping, lymph node suspension from two mice per biological replicate were activated with 0.5 µg/mL anti-mouse CD3 (Biolegend) and 0.5 µg/mL anti-mouse CD28 (eBioscience) in 2×10 mL complete culture medium in six well plates. Proteomics samples were generated in biological triplicate.

For generation of memory-like or effector cytotoxic T lymphocytes (CTL) from mice with polyclonal T cell repertoires, LN or spleen single-cell suspensions at an equal density for WT and Pim dKO cultures (~1–3 million live cells/mL) were activated with 0.5 µg/mL anti-mouse CD3 (Biolegend) and 0.5 µg/mL anti-mouse CD28 (eBioscience) and recombinant human IL-15 (20 ng/mL, Peprotech) or recombinant human IL-2 (20 ng/mL, Proleukin, Novartis) in 10 mL complete culture medium. After 2 days of activation (~40 hr), cells were washed out of activation medium and resuspended at 0.3 million T cells/mL in fresh culture medium and cytokine, 20 ng/mL of IL-15 or IL-2 for memory-like and effector T cells respectively. T cells were subsequently split into fresh medium and cytokine daily at a density of 0.5 million/mL and 0.3 million/mL for memory-like and effector T cells respectively until analysis timepoint.

For generation of effector CTL from TCR-transgenic P14 mice LN single-cell suspensions were activated with 100 ng/mL gp33 peptide, recombinant human IL-2 (20 ng/mL) and recombinant mouse IL12 (2 ng/mL) (Peprotech) for 2 days, before washing out of activation media and splitting cells daily into fresh media containing IL-2 (20 ng/mL) at a density of 0.3 million/mL.

For co-culture experiments Ly5.1+ WT and Pim dKO (Ly5.2+) cell suspensions were mixed in a 50:50 ratio based on total T cell numbers before activation. A portion of the mixed naive T cells were maintained in recombinant mouse IL-7 (5 ng/mL, Peprotech) to verify baseline T cell ratios over time in culture.

For experiments where a pure CD8+ CTL population were required from non-TCR transgenic mice (adoptive transfer, proteomics and RNAseq experiments), CD4 T cells were depleted on day 3 of CTL cultures (i.e. 24 hr after being washed out of initial activation conditions) by negative magnetic selection. T cells were resuspended in MACS buffer (PBS, 1 mM EDTA, 2%FBS) at $10^8$ /mL, blocked with 50 µL/mL Rat Serum, incubated for 10 min with anti-CD4 biotin antibody (5 µg/mL, Biolegend), incubated for 5 min with 125 µL/mL RapidSphere beads (StemCell Technologies), then volume topped up to 2.5 mL and placed in EasySep magnet for 2.5 min. Supernatant containing CD8 T cells was collected and returned to culture as per CTL protocol above. CD4 T cell depletion was confirmed by flow cytometry, with CD4+ cells making up <0.4% of live cells.

In experiments where cytokine was withdrawn T cells were washed at least twice with warm complete medium. Where indicated CTL were treated with 100 nM Tofacitinib (Selleckchem), 20 nM of rapamycin (Merck), 20 ng/mL PDBu (Cell Signaling Technologies) and 500 ng/mL ionomycin (Merck) or the pan PIM kinase inhibitors PIM447 (1 or 5 µM as indicated) and AZD1208 (1 or 10 µM as indicated; both Medchemexpress).

## Flow cytometry

Flow cytometry data was acquired on a FACSVerse using FACSuite software, FACSCanto, or LSR II Fortessa with FACS DIVA software (BD Biosciences) or Novocyte (Acea Biosciences Inc) with NovoExpress software (Agilent). Data was analysed using Flowjo software version 9.9.6 or version 10.6.1 and above (Treestar).

Cell surface staining antibodies conjugated to BV421, BV510, V500, FITC, PE, PerCPCy5.5, PECy7, APC, A647, and APCeF780 were obtained from BD Biosciences, eBioscience, or Biolegend. Fc receptors were blocked using Fc block (BD Biosciences). Antibody staining for surface markers was performed at 1:200 in PBS 1%FBS. Antibody clones were as follows: CD4 (RM4-5), CD8 (53–6.7), CD25 (7D4 or PC61), CD44 (IM7), CD45.1 (A20), CD45.2 (104), CD62L (MEL-14), CD69 (H1.2F), CD71 (RI7217), TCRbeta (H57-597).

For IFNγ and Granzyme B intracellular staining cells were fixed and permeabilised using eBioscience Intracellular Fixation and Permeabilisation kit (eBioscience) as per manufacturer instructions. Cells were stained with anti-IFNγ (XMG1.2) and anti-Granzyme B (NGZB) at 1:100 and 1:200 respectively. For intracellular IFNγ staining, GolgiPlug (BD) was added to the culture 4 hr prior to analysis.

For intracellular phospho S6 staining, cells were fixed with 0.5% paraformaldehyde at 37 °C and permeabilised with ice cold 90% methanol. Cells were stained at 1:100 with anti-phospho S6 ribosomal protein (Ser235/236) Alexa Fluor 647 (D57.2.2E, Cell Signaling Technologies cat#4851 S).

For intracellular phospho-STAT5 staining, cells were fixed with 1% paraformaldehyde at room temperature and permeabilised with ice cold 90% methanol. Cells were stained at 1:200 with rabbit anti-phospho STAT5 (Y694) (C11C5, Cell Signaling Technologies cat#9359 S) followed by anti-rabbit IgG Fab2 Alexa Fluor 647 (Cell Signaling Technologies cat#4414 S).

For intracellular PDCD4 staining, cells were fixed with 1% paraformaldehyde at room temperature and permeabilised with Perm Buffer (eBioscience, cat# 00-8333-56). Cells were stained at 1:100 with rabbit anti-PDCD4 (D29C6, Cell Signaling Technologies cat#9535 S) followed by anti-rabbit IgG Fab2 Alexa Fluor 647 (Cell Signaling Technologies cat#4414 S).

## Cell sorting

Cell sorting was performed on a Sony SH800S cell sorter (Sony Biotechnology). Staining was performed in PBS 1% FBS, and sorting and collection of cells for proteomics analysis was performed in RPMI 1640 containing glutamine, supplemented with 1% FBS. 24 hr TCR activated WT and Pim dKO CD4 and CD8 T cells were sorted as: DAPI-CD69+ CD8+ or CD4+. Cells were washed twice with HBSS (no calcium, no magnesium, no phenol red, GIBCO) before being snap frozen in liquid nitrogen and stored at −80 °C until further processing.

## CTL number estimation and viability

Absolute CD8 T cell number per well/flask and % live cells was assessed in CTL cultures by volumetric measurement of cell number from 50 μL culture medium. Cells were transferred into 450 μL PBS containing low concentration anti-CD8, anti-CD4 antibody (both 1:1500) and DAPI (1 μg/mL, Thermo Fisher Scientific) or propidium iodide (0.2 μg/mL, Sigma) and cell number and viability assessed on either FACSVerse or Novocyte flow cytometers. Estimated live cell counts were corrected for assay dilution ratio, total culture volume and splitting ratios for each time point. Proportions of CD8 T cells that were Ly5.1+ or Ly5.2+ from parallel surface staining were used to calculate WT and Pim dKO numbers respectively in co-culture experiments.

## CTV proliferation assay

For CTV proliferation assay, lymph node suspensions at $10^7$ cells per mL in PBS 0.5%FCS were labelled with 5 μM CTV (Invitrogen) for 20 min at 37 °C before reaction was quenched and cells washed with cold medium. $10^5$ live cells were activated with αCD3/αCD28 (both 0.5 μg/mL) or maintained in IL-7 (5 ng/mL) +/- rapamycin (20 nM) in 200 μL total volume in 96 well flat-bottomed plates. Propidium iodide (0.2 μg/mL), low-dose αCD4, αCD8 antibody (both 1:3200 final concentration) and 10,000 sphero rainbow beads (BD) were added to wells prior to analysis of beads number, cell number, cell viability, CD4 and CD8 T cell composition and CTV profile of samples by flow cytometry. The ratio between the known number of beads added to the well and the number of beads measured per well was multiplied with the number of cells measured to calculate the absolute cell number per well. Cell number per division was then corrected for the effect of expansion by dividing cell number per division by 2^division number and the arithmetic mean division number calculated from these values.

## Protein synthesis assay

To assess protein synthesis cells were cultured with 20 μM O-propargyl puromycin (OPP, Jena Bioscience) for 10 min. Control cells were cultured with cycloheximide (100 μg/mL, Sigma-Aldrich) for 20 min prior to addition of OPP to assess background staining level when cytosolic protein synthesis is completely inhibited. Cells were washed, fixed with 1% PFA, permeabilised with 0.5% Triton x-100 and OPP incorporation into newly synthesised peptides was measured by conjugating with Alexa647-azide via a Click-IT chemistry reaction (2 μM Alexa-647-azide, 2 mM $CuSO_4$, 5 mM ascorbic acid in PBS; Invitrogen) and assessing fluorescence by flow cytometry.

## Adoptive transfer

WT and Pim dKO CD8 T cells were activated and expanded in IL-2 as described above. On day 6 of culture, WT and Pim dKO CTL were labelled with either CTV, or 5 μM CFSE (Invitrogen) as per CTV assay protocol described above, with the exception that labelling was performed in RPMI 1640 and CFSE labelling was for 10 min. CTV and CFSE labelling was alternated between genotypes across biological replicates to prevent any potential introduction of bias due to possible differences in dye toxicity. Labelled WT and Pim dKO CD8 T cells were mixed in a 50:50 ratio and 5 million cells/mouse were adoptively transferred via intravenous tail injection into C57BL/6-recipient mice. Recipient mice were sacrificed and blood, spleen, and lymph nodes (axillary, brachial and inguinal) removed 4 and 23 hr post-transfer and the ratio of CTV to CFSE cells assessed by flow cytometry. Suitable group sizes were determined based on previous experience with this experimental system (*Sinclair et al., 2008*; *Finlay et al., 2012*). If fewer than 100 transferred cells were recovered in the flow cytometry analysis of organs then quantifications were considered to be of insufficient confidence and were not plotted.

## Western blotting

Cell pellets were lysed in RIPA buffer (100 mM HEPES, pH 7.4, 150 mM NaCl, 1% NP40, 0.1% SDS, 0.5% sodium deoxycholate, 10% glycerol, 1 mM EDTA, 1 mM EGTA, 1 mM TCEP, and protease and phosphatase inhibitors (Roche)) at 20 million T cells/mL at 4 °C. Lysates were sonicated, centrifuged 12 min, 13,000 rpm, 4 °C. Supernatant was transferred to fresh tube and NuPAGE LDS sample buffer (1×) (Life Technologies) and tris(2-carboxyethyl)phosphine (TCEP, 25 mM, Thermo Fisher) was added before boiling samples at 100 °C for 3 min. Samples were loaded and separated by SDS-PAGE (NuPAGE 4–12% gradient precast gels (Thermo Fisher) or 12% polyacrylamide running gel), and then transferred to nitrocellulose membranes (Whatman). Equal cell numbers (~140,000 cells) were loaded per

lane. Blots were probed with the following primary antibodies: S6K (5G10, Cell Signaling Technology, cat# 2217 S), S6K p-T389 (108D2, Cell Signaling Technology, cat# 9234 S), STAT5 p-Y694 (C11C5, Cell Signalling Technology, cat# 9359 S), Pim1 (12H8, Santa Cruz, cat# SC-13513), Pim2 (1D12, Santa Cruz, cat# SC-13514). Horseradish peroxidase (HRP)-conjugated anti-rabbit or anti-mouse secondary antibodies were used to assess protein signal (Thermo Scientific, Cell Signaling Technologies). After 1 min treatment with Immobilon Western Chemiluminscent HRP Substrate (Merck Millipore) chemiluminescence (2 min exposure time) of membranes was measured using an Odyssey Fc Imaging System (Licor). Western blot images in *Figure 6A* have been reflected horizontally in Adobe Illustrator 2021 for ease of visualisation. For quantification in *Figure 6A*, Image Studio Software (LI-COR) was used to measure chemiluminescence signal on raw image files.

## Proteomics

### Proteomics sample preparation

Cell pellets for 24 hr TCR WT and Pim dKO CD4 and CD8 T cells were activated and sorted as described above.

IL-2 and IL-15 expanded WT and Pim dKO CD8 T cells on day 6 of culture days, were cultured in fresh warm complete medium and cytokine for 2 hr before cells were harvested, washed twice in ice-cold HBSS and cell pellets snap frozen in liquid nitrogen. Duplicate cell pellets were collected per sample for parallel processing for proteomics or RNAseq analysis.

IL-2 expanded P14 CD8 T cells on day 5 of culture were treated for 24 hr with PIM447 (5 µM) or AZD1208 (10 µM). Cells were harvested on day 6 of culture, washed twice in ice-cold HBSS and cell pellets snap frozen in liquid nitrogen.

For all proteomics samples, cell pellets were lysed in 4% SDS, 50 mM TEAB pH 8.5, 10 mM TCEP (5 min, 1200 rpm, room temperature), boiled (5 min, 500 rpm, 95 °C), then sonicated with a BioRuptor (30 s on, 30 s off x30 cycles). Protein concentration was determined using EZQ protein quantitation kit (Invitrogen) as per manufacturer instructions. Lysates were alkylated with 20 mM iodoacetamide (Sigma) for 1 hr at room temperature in the dark.

24 hr TCR WT and Pim dKO CD4 and CD8 T cell proteomics samples and Day 6, 24 hr pan-PIM kinase inhibitor treated IL-2 expanded CD8 T cell samples were cleaned up and peptides generated by an SP3 bead protocol (*Hughes et al., 2014*) as described previously (*Marchingo et al., 2020*). As outlined in reference (*Marchingo et al., 2020*): "Briefly, 200 µg of 1:1 mixed Hydrophobic and Hydrophilic Sera-Mag SpeedBead Carboxylate-Modified Magnetic Particles were added per protein sample then acidified to ~pH 2.0 by addition 10:1 Acetonitrile: Formic Acid. Beads were immobilised on a magnetic rack and proteins washed with 2×70% ethanol and 1×100% acetonitrile. Rinsed beads were reconstituted in 0.1% SDS 50 mM TEAB pH 8.5, 1 mM CaCl2 and digested overnight with LysC followed by overnight digestion with Trypsin, each at a 1:50 enzyme to protein ratio. Peptide clean up was performed as per SP3 procedure (*Hughes et al., 2014*). Briefly, protein-bead mixtures were resuspended and 100% acetonitrile added for 10 min (for the last 2 min of this beads were immobilised on a magnetic rack). Acetonitrile and digest buffer were removed, peptides were washed with acetonitrile and eluted in 2% DMSO. Peptide concentration was quantified using CBQCA protein quantitation kit (Invitrogen) as per manufacturer protocol. Formic acid was added to 5% final concentration".

24 hr TCR WT and Pim dKO CD4 and CD8 T cell proteomics samples were fractionated using high pH reverse phase liquid chromatography as described previously (*Marchingo et al., 2020*). As outlined in reference (*Marchingo et al., 2020*): "Samples were loaded onto a 2.1 mm x 150 mm XBridge Peptide BEH C18 column with 3.5 µm particles (Waters). Using a Dionex Ultimate3000 system, the samples were separated using a 25 min multistep gradient of solvents A (10 mM formate at pH 9 in 2% acetonitrile) and B (10 mM ammonium formate pH 9 in 80% acetonitrile), at a flow rate of 0.3 mL/min. Peptides were separated into 16 fractions which were consolidated into eight fractions. Fractionated peptides were dried in vacuo then dissolved in 5% Formic Acid for analysis by LC-ES-MS/MS".

24 hr pan-PIM kinase inhibitor treated day 6 IL-2 expanded CD8 T cell samples, were not fractionated. They were dried in vacuo then dissolved in 5% Formic Acid for analysis by LC-ES-MS/MS after SP3 cleanup.

Day 6 IL-2 and IL-15 expanded WT and PimdKO CD8 T cells samples were cleaned up and peptides generated by an S-Trap protocol (*Zougman et al., 2014*) as per manufacturer protocol (Protifi). Briefly, add phosphoric acid (to final concentration 1.2%) then S-Trap binding buffer (90% methanol,

100 mM TEAB, pH 7.1) at a ratio 1:7 v:v lysate:binding buffer was added to sample lysates containing 100–300 µg total protein. Acidified lysates were loaded to S-Trap mini columns and spun (4000 ×$g$, 30 s) until all SDS lysate/S-Trap buffer had passed through the column. Proteins were digested on column in digest buffer (50 mM sodium bicarbonate) containing Trypsin at a 1:20 enzyme to protein ratio (2 hours, 47 °C without shaking). Peptides were eluted by spinning in digest buffer (1000 g, 1 min), then 0.2% aqueous formic acid (1000 g, 1 min), then 50% acetonitrile 0.2% formic acid (4000 g, 1 min). Peptide concentration was quantified using CBQCA protein quantitation kit (Invitrogen) as per manufacturer protocol. Peptides were dried in vacuo then dissolved in 5% Formic Acid for analysis by LC-ES-MS/MS.

## Liquid chromatography electrospray tandem mass spectrometry analysis (LC-ES-MS/MS)

≤1 µg of peptide was analysed per fraction by Data-Dependent Acquisition (DDA) Mass Spectrometry for 24 hr TCR WT and Pim dKO CD4 and CD8 T cell proteomics samples. For label-free DDA proteomics of WT and Pim dKO CD4 and CD8 24 hr TCR activated T cells samples were analysed as described previously (*Sinclair et al., 2019*). As outlined in reference (*Sinclair et al., 2019*): "samples were injected onto a nanoscale C18 reverse-phase chromatography system (UltiMate 3000 RSLC nano, Thermo Scientific) then electrosprayed into an Orbitrap mass spectrometer (LTQ Orbitrap Velos Pro; Thermo Scientific). For chromatography buffers were as follows: HPLC buffer A (0.1% formic acid), HPLC buffer B (80% acetonitrile and 0.08% formic acid) and HPLC buffer C (0.1% formic acid). Peptides were loaded onto an Acclaim PepMap100 nanoViper C18 trap column (100 µm inner diameter, 2 cm; Thermo Scientific) in HPLC buffer C with a constant flow of 10 µL/min. After trap enrichment, peptides were eluted onto an EASY-Spray PepMap RSLC nanoViper, C18, 2 µm, 100 Å column (75 µm, 50 cm; Thermo Scientific) using the buffer gradient: 2% B (0–6 min), 2% to 35% B (6–130 min), 35% to 98% B (130–132 min), 98% B (132–152 min), 98% to 2% B (152–153 min), and equilibrated in 2% B (153–170 min) at a flow rate of 0.3 µl/min. The eluting peptide solution was automatically electrosprayed using an EASY-Spray nanoelectrospray ion source at 50° and a source voltage of 1.9 kV (Thermo Scientific) into the Orbitrap mass spectrometer (LTQ Orbitrap Velos Pro; Thermo Scientific). The mass spectrometer was operated in positive ion mode. Full-scan MS survey spectra (mass/charge ratio, 335–1800) in profile mode were acquired in the Orbitrap with a resolution of 60,000. Data were collected using data-dependent acquisition: the 15 most intense peptide ions from the preview scan in the Orbitrap were fragmented by collision-induced dissociation (normalized collision energy, 35%; activation Q, 0.250; activation time, 10ms) in the LTQ after the accumulation of 5000 ions. Precursor ion charge state screening was enabled, and all unassigned charge states as well as singly charged species were rejected. The lock mass option was enabled for survey scans to improve mass accuracy. (Using Lock Mass of 445.120024)".

~0.2 µg of peptide was analysed per sample by single shot DDA Mass Spectrometry for 24 hr pan-PIM kinase inhibitor treated, day 6 IL-2 expanded CD8 T cell samples.

Samples were injected onto a nanoscale C18 reverse-phase chromatography system (UltiMate 3000 RSLC nano, Thermo Fisher Scientific) then electrosprayed into a Q-Exactive HF (Thermo Fisher Scientific). A 2–35% B gradient comprising of eluent A (0.1% formic acid) and eluent B (80% acetonitrile/0.1% formic acid) was used to run a 120 minute gradient per sample. With the mass spectrometer in positive mode the top 20 most intense peaks from a mass range of 335–1800 m/z in each MS1 scan with a resolution of 60,000 were then taken for MS2 analysis at a resolution of 15,000. Spectra were fragmented using Higher-energy C-trap dissociation (HCD).

2 µg of peptide was analysed per sample by Data-Independent Acquisition (DIA) Mass Spectrometry for the IL-2 and IL-15 expanded CD8 T cell proteomics samples. For label-free DIA proteomics of Day 6 IL-2 and IL-15 expanded WT and Pim dKO CD8 T cells peptide samples were analysed as described in *Sollberger et al., 2023* with some minor differences. Peptides were injected onto a nanoscale C18 reverse-phase chromatography system (UltiMate 3000 RSLC nano, Thermo Scientific) then electrosprayed into an Orbitrap Exploris 480 Mass Spectrometer (Thermo Scientific). For liquid chromatography buffers were as follows: buffer A (0.1% formic acid in Milli-Q water (v/v)) and buffer B (80% acetonitrile and 0.1% formic acid in Milli-Q water (v/v)). Sample were loaded at 10 µL/min onto a trap column (100 µm×2 cm, PepMap nanoViper C18 column, 5 µm, 100 Å, Thermo Fisher Scientific) equilibrated in 0.1% trifluoroacetic acid (TFA). The trap column was washed for 3 min at the

same flow rate with 0.1% TFA then switched in-line with a Thermo Scientific, resolving C18 column (75 μm×50 cm, PepMap RSLC C18 column, 2 μm, 100 Å). The peptides were eluted from the column at a constant flow rate of 300 nL/min with a linear gradient from 3% buffer B to 6% buffer B in 5 min, then from 6% buffer B to 35% buffer B in 115 min, and finally to 80% buffer B within 7 min. The column was then washed with 80% buffer B for 4 min and re-equilibrated in 3% buffer B for 15 min. Two blanks were run between each sample to reduce carry-over. The column was kept at a constant temperature of 40 °C.

The data was acquired using an easy spray source operated in positive mode with spray voltage at 2.1 kV and the ion transfer tube temperature at 275 °C. The MS was operated in DIA mode. A scan cycle comprised a full MS scan m/z range from 350 to 1650, with RF lens at 40%, AGC target set to custom, normalised AGC target at 300%, maximum injection time mode set to custom, a maximum ion injection time of 20ms, microscan set to 1 and source fragmentation disabled. MS survey scan was followed by MS/MS DIA scan events using the following parameters: multiplex ions set to false, collision energy mode set to stepped, collision energy type set to normalized, HCD collision energies set to 25.5, 27, and 30%, orbitrap resolution 30000, first mass 200, RF lens 40%, AGC target set to custom, normalized AGC target 3000%, microscan set to 1 and maximum injection time 55ms. Data for both MS scan and MS/MS DIA scan events were acquired in profile mode.

## Proteomics data analysis

The 24 hr TCR WT and Pim dKO CD4 and CD8 T cell DDA proteomics data and 24 hr pan-PIM kinase inhibitor treated Day 6 IL-2 expanded CD8 T cell DDA proteomics data were processed, searched, and quantified with the MaxQuant software package, version 1.6.10.43. For the protein and peptide searches, we generated a hybrid database consisting of all manually annotated mouse SwissProt entries, combined with mouse TrEMBL entries with protein level evidence available and a manually annotated homologue within the human SwissProt database as described in *Marchingo et al., 2020*. For the 24 hr TCR activated data, this was from the Uniprot release 2020_06, for the 24 hr PIM kinase inhibitor data this was from the Uniprot release 2019_08. Search parameters were as described in *Marchingo et al., 2020*: "The following MaxQuant search parameters were used: protein N-terminal acetylation, methionine oxidation, glutamine to pyroglutamate, and glutamine and asparagine deamidation were set as variable modifications and carbamidomethylation of cysteine residues was selected as a fixed modification; Trypsin and LysC were selected as the enzymes with up to two missed cleavages permitted; the protein and PSM false discovery rate was set to 1%; matching of peptides between runs was switched off".

Data filtering and protein copy number quantification was performed in the Perseus software package, version 1.6.6.0. Proteins were quantified from unique peptides and razor peptides (peptides assigned to a group, but not unique to that group). The data set was filtered to remove proteins categorised as 'contaminants', 'reverse' and 'only identified by site'.

To generate a point of reference between TCR activated WT and Pim dKO and inactive T cells, naive WT CD4 and CD8 T cell data from *Marchingo et al., 2020*, which was collected and processed identically, was included in proteomics search, results table and plot of protein content. At this point biological replicate 3 from Pim dKO CD4 T cells was excluded from further analysis due to poor sample quality (e.g. protein sequence coverage was an average of 13.9% compared to an average of 18.2–24.6% for all other samples and estimated total protein content per cell was ~1/3rd of that observed in the other two biological replicates for that condition, despite no obvious difference in FSC measured when FACS sorting).

The day 6 IL-2 and IL-15 expanded WT and Pim dKO CD8 T cell DIA proteomics data files were searched using Spectronaut version 14.7. Data was analysed using a library-free approach. Raw proteomics data files were searched using the Pulsar tool within Spectronaut using the following settings: 0.01 FDR at the protein and peptide level with digest rule set to 'Trypsin'P'. A maximum of two missed cleavages and minimum peptide length of 7 amino acids was selected. Carbamidomethyl of cysteine was selected as a fixed modification while protein n-terminal acetylation and methionine oxidation were selected as variable modifications. Manufacturer default settings were used for identification, with protein and precursor q-value set to 0.01 using default decoy strategy. For quantification settings, the MS-Level quantity was set to 'MS2', major group Top N and minor group Top N were

set as 'False' and profiling was set as 'False'. The data was searched against the same mouse Uniprot database as used above for DDA analysis.

Data was filtered to only include proteins for which at least one condition had peptides detected in ≥2 biological replicates for 24 hr TCR and day 6 IL-2 and IL-15 WT and Pim dKO data. Data was filtered to only include proteins for which at least two conditions had peptides detected in ≥2 biological replicates for PIM kinase inhibitor data. Mean copy number per cell was calculated using the 'proteomic ruler' plugin as described in *Wiśniewski et al., 2014*.

Mass contribution of proteins (g/cell) was calculated as (protein copy number) * (molecular weight (Daltons)) / (Avogadro's constant). Differential expression analysis of protein copy number was performed using RStudio (version 1.2.5033). p-values and fold-change were calculated with the Bioconductor package Limma (version 3.44.3). Q-values were calculated using Bioconductor package qvalue (version 2.20.0). All scripts used to perform analysis are available from authors upon request.

## RNAseq
### RNA extraction, library preparation and RNA sequencing
Biological triplicate cell pellets of day 6 IL-2 or IL-15 differentiated CD8 T cells were collected in parallel with proteomics samples as described above. Total RNA was extracted from cell pellets with Qiagen RNAeasy mini kit (Qiagen) as per manufacturer instructions.

2 µg of RNA was submitted per sample to the Finnish Functional Genomics Centre, University of Turku and Åbo Akademi University for library preparation and sequencing. RNA quality was verified using Agilent Bioanalyzer 2100 or Advanced Analytical Fragment Analyzer. 300 ng of total RNA was used for library preparation using the Illumina TruSeq Stranded mRNA sample preparation kit (15031047) as per manufacturer instructions. Library quality was confirmed with Advanced Analytical Fragment Analyzer and quantified with Qubit Fluorometric Quantitation (Life Technologies). Sample libraries were pooled and run in a single lane on an Illumina NovaSeq 6000 with read length 2x50 bp. Base calling was performed with bcl2fastq2 conversion software (NovaSeq 6000) and automatic adapter trimming performed for fastq raw read files generated.

### RNAseq data analysis
Quality assessment of raw reads was evaluated using MultiQC (*Ewels et al., 2016*). Raw reads were aligned to the GRCm38.p6 (Ensembl) build of the *Mus musculus* reference genome using the STAR aligner (*Dobin et al., 2013*; version 2.7.1 a), with 'outFilterType BySJout' option switched on to reduce spurious junctions. The STAR quantMode GeneCounts option was used to map aligned reads to genes to generate a gene counts matrix, using the annotation file GRCm38.101 (Ensembl).

Bioconductor package EdgeR (version 3.30.3) (*Robinson et al., 2010*; *McCarthy et al., 2012*) was used to reduce noise from low count genes with filterByExpr function and the resulting count matrix was normalised by the EdgeR Trimmed Mean of M-values (TMM) method.

Differential expression analysis was performed using Limma package (version 3.44.3) (*Ritchie et al., 2015*; *Law et al., 2016*). Reads were converted to $\log_2$ counts per million and weighted using the voomWithQualityWeights function. A linear model was fitted to each gene and empirical Bayes moderated t-statistics were applied. q-values were calculated from using Bioconductor package qvalue (version 2.20.0) (*Storey et al., 2023*). The calculateTPM fuction from the Bioconductor package Scater (version 1.16.2) (*McCarthy et al., 2017*) was used to calculate TPM values for visualisation purposes, with total exon length calculated using GenomicRanges (version 1.40.0) (*Lawrence et al., 2013*) as the sum of the non-overlapping exonic regions for a gene.

## Combining proteomics and RNAseq data
Proteomics data was annotated during Spectronaut search and Perseus analysis with Uniprot ID, MGI symbol and MGI ID information. BioMart (version 2.47.2) was used to extract UniProt identifiers, MGI symbols and MGI IDs for corresponding ENMUSG IDs of RNAseq data. RNAseq data was first combined with proteomics data based on UniProt ID. Proteins that didn't receive an initial match, were then combined by matching MGI ID values, then if still unmatched by MGI symbol. Data combinations were manually checked for obvious mis-annotations. Proteins with no corresponding mRNA identified or vice versa were excluded from the combined data file. In a few instances, multiple proteins aligned to a single mRNA measurement or multiple mRNA aligned to a single protein measurement. For the

purposes of this broad examination of how differences in expression at the protein or mRNA level these results were also excluded from the combined data table and subsequent analysis.

## Graphing and statistics

Heatmaps were generated using Broad Institute software Morpheus (https://software.broadinstitute.org/morpheus). Statistical tests performed for experiments other than proteomics and RNAseq are indicated in figure legends and were performed in Prism (GraphPad, version 9 or 10).

## Acknowledgements

The authors thank members of the Cantrell group for their critical discussion of the data, in particular we thank Linda Sinclair for critical reading and comments on the manuscript. We thank Victor Tybulewicz for providing the Pim1-/- and Pim2-/- mice, E Emslie for help with mouse maintenance and experimental support, B Stubbs for technical assistance with mouse genotyping, A Brenes for assistance with analytical tools, A Gardner, A Rennie and R Clarke from the Flow Cytometry facility for cell sorting, K Beattie and A Atrih from the Fingerprint proteomics facility for performed mass spectrometry, S Thomson for assistance with adoptive transfer experiments, the Biological Resource Unit at the University of Dundee and the Finnish Functional Genomics Centre, University of Turku and Abo Akademi Biocenter Finland for performing RNA sequencing. This research was supported by a Wellcome Trust Principal Research Fellowship to DAC (205023/Z/16/Z), a Wellcome Trust Equipment Award to DAC (202950/Z/16/Z), an EMBO Long term Fellowship (ALTF 1543–2015) and NHMRC CJ Martin Early Career fellowship (APP1120748) to JMM and has received funding from the European Union's Horizon 2020 research and innovation programme under the Marie Skłodowska-Curie grant agreement No 705984 to JMM and DAC.

## Additional information

### Funding

| Funder | Grant reference number | Author |
|---|---|---|
| Wellcome Trust | 10.35802/205023 | Doreen A Cantrell |
| Wellcome Trust | 10.35802/202950 | Doreen A Cantrell |
| European Molecular Biology Organization | ALTF 1543-2015 | Julia M Marchingo |
| National Health and Medical Research Council | APP1120748 | Julia M Marchingo |
| Horizon 2020 Framework Programme | 10.3030/705984 | Julia M Marchingo Doreen A Cantrell |

The funders had no role in study design, data collection and interpretation, or the decision to submit the work for publication. For the purpose of Open Access, the authors have applied a CC BY public copyright license to any Author Accepted Manuscript version arising from this submission.

### Author contributions

Julia M Marchingo, Conceptualization, Data curation, Formal analysis, Funding acquisition, Investigation, Visualization, Methodology, Writing – original draft, Project administration, Writing – review and editing; Laura Spinelli, Investigation, Visualization, Methodology, Writing – review and editing; Shalini Pathak, Methodology, Writing – review and editing; Doreen A Cantrell, Conceptualization, Resources, Supervision, Writing – original draft, Project administration, Writing – review and editing

### Author ORCIDs

Julia M Marchingo https://orcid.org/0000-0001-8823-9718
Laura Spinelli https://orcid.org/0000-0002-5801-6297
Doreen A Cantrell https://orcid.org/0000-0001-7525-3350

### Ethics

All animal experiments were performed under Project License PPL 60/4488 and P4BD0CE74.The University of Dundee Welfare and Ethical Use of Animals Committee accepted the project licence for submission to the UK Home Office. All studies, breeding and maintenance performed in Dundee in compliance with UK Home Office Animals (Scientific Procedures) Act 1986 guidelines. Individual study plans were approved and deemed compliant by the UVS/Named Compliance Officer.

Reviewer #1 (Public review): https://doi.org/10.7554/eLife.98622.3.sa1
Reviewer #2 (Public review): https://doi.org/10.7554/eLife.98622.3.sa2
Author response https://doi.org/10.7554/eLife.98622.3.sa3

---

## Additional files

### Supplementary files

Supplementary file 1. Protein copy numbers and statistical test results for 24 hour TCR-activated WT and Pim dKO CD4 and CD8 T cells proteomics data.

Supplementary file 2. mRNA expression data, mRNA annotation and statistical test results for RNAseq analysis of Day 6 IL-15 expanded WT and Pim dKO CD8 T cells.

Supplementary file 3. Protein copy numbers and statistical test results for proteomics analysis of Day 6 IL-15 expanded WT and Pim dKO CD8 T cells.

Supplementary file 4. Protein copy numbers and statistical test results for proteomics analysis of Day 6 IL-2 expanded WT and Pim dKO CD8 T cells.

Supplementary file 5. Protein copy numbers and statistical test results for proteomics analysis of Day 6 IL-2 expanded P14 CD8 T cells treated for 24 hours with pan PIM kinase inhibitors PIM447 or AZD1208.

Supplementary file 6. mRNA expression data and statistical test results for RNAseq analysis of Day 6 IL-2 expanded WT and Pim dKO CD8 T cells.

Supplementary file 7. Combined proteomics and RNAseq expression data and statistical test results from Day 6 IL-2 expanded WT and Pim dKO CD8 T cells.

MDAR checklist

### Data availability

All data generated or analysed during this study are included in the manuscript and supporting files. Raw mass spec data files and MaxQuant or Spectronaut analysis files are available on the ProteomeXchange data repository. Datasets generated for this study can be accessed with the following identifiers: PXD051198 for 24 hour TCR activated WT and Pim dKO CD4 and CD8 T cells; Naïve T cell data used as a point of reference for plotting 24 hour TCR proteomics has previously been published as part of *Marchingo et al., 2020* and can also be found under the identifier PXD016105; PXD051214 for Pim inhibitor proteomics; PXD051213 for day 6 IL-2 and IL-15 expanded WT and Pim dKO CD8 T cells. An easy-to-use graphical interface for examining protein copy number expression from the 24-hour TCR WT and Pim dKO CD4 and CD8 T cell proteomics and IL-2 and IL-15 expanded WT and Pim dKO CD8 T cell proteomics datasets is also available on the Immunological Proteome Resource website: immpres.co.uk (*Brenes et al., 2023*) under the Cell type(s) selection: "T cell specific" and Dataset selection: "Pim1/2 regulated TCR proteomes" and "Pim1/2 regulated IL2 or IL15 CD8 T cell proteomes". Raw RNAseq data files are available on the NCBI GEO website and can be accessed with the identifier GSE261667. Processed data for quantification and statistical analysis of newly generated proteomics and RNAseq experiments have been provided as *Supplementary files 1–7*.

The following datasets were generated:

| Author(s) | Year | Dataset title | Dataset URL | Database and Identifier |
|---|---|---|---|---|
| Marchingo JM, Spinelli L, Pathak S, Cantrell DA | 2024 | Proteomics of 24 h TCR activated wild-type and Pim1/Pim2 double knockout CD4 and CD8 T cells | https://www.ebi.ac.uk/pride/archive/projects/PXD051198 | PRIDE, PXD051198 |
| Marchingo JM, Spinelli L, Pathak S, Cantrell DA | 2024 | Whole Proteomics of 24 hr pan Pim kinase inhibitor treatment of IL2 cultured CD8 T cells | https://www.ebi.ac.uk/pride/archive/projects/PXD051214 | PRIDE, PXD051214 |
| Marchingo JM, Spinelli L, Pathak S, Cantrell DA | 2024 | Proteomics of Day 6 IL-2 or IL-15 expanded wild-type and Pim1/Pim2 double knockout CD8+ T cells | https://www.ebi.ac.uk/pride/archive/projects/PXD051213 | PRIDE, PXD051213 |
| Marchingo JM, Spinelli L, Emslie E, Pathak S, Cantrell DA | 2025 | RNAseq of Pim1/Pim2 double knockout vs wild-type CD8+ T cells expanded in IL2 or IL15 | https://www.ncbi.nlm.nih.gov/geo/query/acc.cgi?acc=GSE261667 | NCBI Gene Expression Omnibus, GSE261667 |

The following previously published datasets were used:

| Author(s) | Year | Dataset title | Dataset URL | Database and Identifier |
|---|---|---|---|---|
| Marchingo JM, Sinclair LV, Howden AJM, Cantrell DA | 2020 | Proteome of naive and TCR activated wild-type, Myc-deficient and Slc7a5-deficient T cells | https://www.ebi.ac.uk/pride/archive/projects/PXD016105 | PRIDE, PXD016105 |
| Marchingo JM, Sinclair LV, Howden AJM, Cantrell DA | 2020 | OT1 T cell activation time course | https://www.ebi.ac.uk/pride/archive/projects/PXD016443 | PRIDE, PXD016443 |
| Spinelli L, Marchingo JM, Nomura A, Damasio MP, Cantrell DA | 2021 | RNAseq data for ex vivo Naïve CD8+ T cells and TCR activated 24hr in the presence or absence of PI3Kp110δ inhibitor | https://www.ncbi.nlm.nih.gov/geo/query/acc.cgi?acc=GSE169436 | NCBI Gene Expression Omnibus, GSE169436 |
| Brenes AJ, Cantrell DA | 2019 | Defining the T cell proteome and the role of mTORC1 during differentiation using quantitative proteomics | https://www.ebi.ac.uk/pride/archive/projects/PXD012058 | PRIDE, PXD012058 |
| Brenes AJ, Cantrell DA | 2020 | Immunological Proteome Resource (ImmPRes): Part two | https://www.ebi.ac.uk/pride/archive/projects/PXD020091 | PRIDE, PXD020091 |

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
