## [Editor Report · eLife Assessment]

These **important** findings detail the role of Pim1 and Pim2 in controlling the behaviour and activity of 'killer' T cells; a vital cell within of our immune system. The authors capitalized on high resolution quantitative analysis of the proteomes and transcriptomes of Pim1/Pim2-deficient CD8 T cells to provide **compelling** evidence for how the PIM1/2 kinases control TCR-driven activation and IL-2/IL-15-driven proliferation and differentiation into effector T cells. It's also noteworthy that Pim1/Pim2 impact is better revealed through quantitative proteomics than transcriptomics.

---

## [Referee Report · Reviewer #1 (Public review)]

Summary and Strengths:

The study focuses on PIM1 and 2 in CD8 T cell activation and differentiation. These two serine/threonine kinases belong to a large network of Serine/Threonine kinases that acts following engagement of the TCR and of cytokine receptors and phosphorylates proteins that control transcriptional, translational and metabolic programs that result in effector and memory T cell differentiation. The expression of PIM1 and PIM2 is induced by the T-cell receptor and several cytokine receptors. The present study capitalized on high-resolution quantitative analysis of the proteomes and transcriptomes of Pim1/Pim2-deficient CD8 T cells to decipher how the PIM1/2 kinases control TCR-driven activation and IL-2/IL-15-driven proliferation, and differentiation into effector T cells.

Quantitative mass spectrometry-based proteomics analysis of naive OT1 CD8 T cell stimulated with their cognate peptide showed that the PIM1 protein was induced within 3 hours of TCR engagement and its expression was sustained at least up to 24 hours. The kinetics of PIM2 expression was protracted as compared to that of PIM1. Such TCR-dependent expression of PIM1/2 correlated with the analysis of both Pim1 and Pim2 mRNA. In contrast, Pim3 mRNA was only expressed at very low levels and the PIM3 protein not detected by mass spectrometry. Therefore, PIM1 and 2 are the major PIM kinases in recently activated T cells. Pim1/Pim2 double knockout (Pim dKO) mice were generated on a B6 background and found to express lower number of splenocytes. No difference in TCR/CD28-driven proliferation was observed between WT and Pim dKO T cells over 3 days in culture. Quantitative proteomics of >7000 proteins further revealed no substantial quantitative or qualitative differences in protein content or proteome composition. Therefore, other signaling pathways can compensate for the lack of PIM kinases downstream of TCR activation.

Considering that PIM1 and PIM2 kinase expression is regulated by IL-2 and IL-15, antigen-primed CD8 T cells were expanded in IL-15 to generate memory phenotype CD8 T cells or expanded in IL-2 to generate effector cytotoxic T lymphocytes (CTL). Analysis of the survival, proliferation, proteome, and transcriptome of Pim dKO CD8 T cells kept for 6 days in IL-15 showed that PIM1 and PIM2 are dispensable to drive the IL-15-mediated metabolic or differentiation programs of antigen-primed CD8 T cells. Moreover, Pim1/Pim2-deficiency had no impact on the ability of IL-2 to maintain CD8 T cell viability and proliferation. However, WT CTL downregulated expression of CD62L whereas the Pim dKO CTL sustained higher CD62L expression. Pim dKO CTL were also smaller and less granular than WT CTL. Comparison of the proteome of day 6 IL-2 cultured WT and Pim dKO CTL showed that the latter expressed lower levels of the glucose transporters, SLC2A1 and SLC2A3, of a number of proteins involved in fatty acid and cholesterol biosynthesis, and CTL effector proteins such as granzymes, perforin, IFNg and TNFa. Parallel transcriptomics analysis showed that the reduced expression of perforin and some granzymes correlated with a decrease in their mRNA whereas the decreased protein levels of granzymes B and A, and of the glucose transporters SLC2A1 and SLC2A3 did not correspond with decreased mRNA expression. Therefore, PIM kinases are likely required for IL-2 to maximally control protein synthesis in CD8 CTL. Along that line, the translational repressor PDCD4 was increased in Pim dKO CTL and pan-PIM kinase inhibitors caused a reduction in protein synthesis rates in IL-2 expanded CTL. Finally, the differences between Pim dKO and WT CTL in terms of CD62L expression resulted in that Pim dKO CTL but not WT CTL retained the capacity to home to secondary lymphoid organs. In conclusion, this thorough and solid study showed that the PIM1/2 kinases shape the effector CD8 T cell proteomes rather than transcriptomes and are important mediators of IL2-signalling and CD8 T cell trafficking.

Weaknesses: None

Comments on revisions:

The authors have been able to provide in their rebuttal letter fair answers to most of the queries primarily raised by Reviewer 2 and they have incorporated the corresponding results in the revised text. It makes the paper stronger.

---

## [Referee Report · Reviewer #2 (Public review)]

Summary:

Using a suite of techniques (e.g., RNA seq, proteomics, and functional experiments ex vivo) this paper extensively focuses on the role of PIM1/2 kinases during CD8 T-cell activation and cytokine-driven (i.e., IL-2 or IL-15) differentiation. The authors key finding is that PIM1/2 enhance protein synthesis in response to IL-2 stimulation, but not IL-15, in CD8+ T cells. Loss of PIM1/2 made T cells less 'effector-like', with lower granzyme and cytokine production, and a surface profile that maintained homing towards secondary lymphoid tissue. The cytokines the authors focus on are IL-15 and Il-2, which drive naïve CD8 T cells towards memory or effector states, respectively. Although PIM1/2 are upregulated in response to T-cell activation and cytokine stimulation (e.g., IL-15, and to a greater extent, IL-2), using T cells isolated from a global mouse genetic knockout background of PIM1/2, the authors find that PIM1/2 did not significantly influence T-cell activation, proliferation, or expression of anything in the proteome under anti-CD3/CD28 driven activation with/without cytokine (i.e., IL-15) stimulation ex vivo. This is perhaps somewhat surprising given PIM1/2 are upregulated, albeit to a small degree, in response to IL-15, and yet PIM1/2 did not seem to influence CD8+ T cell differentiation towards a memory state. Even more surprising is that IL-15 was previously shown to influence the metabolic programming of intestinal intraepithelial lymphocytes, suggesting cell-type specific effects from PIM kinases. What the authors went on to show, however, is that PIM1/2 KO altered CD8 T cell proteomes in response to IL-2. Using proteomics, they saw increased expression of homing receptors (i.e., L-selectin, CCR7), but reduced expression of metabolism-related proteins (e.g., GLUT1/3 & cholesterol biosynthesis) and effector-function related proteins (e.g., IFNy and granzymes). Rather neatly, by performing both RNA-seq and proteomics on the same IL-2 stimulated WT vs. PIM1/2 KO cells, the authors found that changes at the proteome level were not corroborated by differences in RNA uncovering that PIM1/2 predominantly influence protein synthesis/translation. Effectively, PIM1/2 knockout reduced the differentiation of CD8+ T cells towards an effector state. In vivo adoptive transfer experiments showed that PIM1/2KO cells homed better to secondary lymphoid tissue, presumably owing to their heightened L-selectin expression (although this was not directly examined).

Strengths:

Overall, I think the paper is scientifically good, and I have no major qualms with the paper. The paper as it stands is solid, and while the experimental aim of this paper was quite specific/niche, it is overall a nice addition to our understanding of how serine/threonine kinases impact T cell state, tissue homing, and functionality. Of note, they hint towards a more general finding that kinases may have distinct behaviour in different T-cell subtypes/states. I particularly liked their use of matched RNA-seq and proteomics to first suggest that PIM1/2 kinases may predominantly influence translation (then going on to verify this via their protein translation experiment - although I must add this was only done using PIM kinase inhibitors not the PIM1/2KO cells). I also liked that they used small molecule inhibitors to acutely reduce PIM1/2 activity, which corroborated some of their mouse knockout findings - this experiment helps resolve any findings resulting from potential adaptation issues from the PIM1/2 global knockout in mice but also gives it a more translational link given the potential use of PIM kinase inhibitors in the clinic. The proteomics and RNA seq dataset may be of general use to the community, particularly for analysis of IL-15 or IL-2 stimulated CD8+ T cells.

Weaknesses:

None. My comments here have been addressed in the previous review.

---

## [Author Response]

The following is the authors’ response to the original reviews

We thank the reviewers for their careful reading of our manuscript and their considered feedback. Please see our detailed response to reviewer comments inset below.

In addition to requested modifications we have also uploaded the proteomics data from 2 of the experiments contained within the manuscript onto the Immunological Proteome Resource (ImmPRes) website: immpres.co.uk making the data available in an easy-to-use graphical format for interested readers to interrogate and explore. We have added the following text to the data availability section (lines 1085-1091) to indicate this:

“An easy-to-use graphical interface for examining protein copy number expression from the 24-hour TCR WT and Pim dKO CD4 and CD8 T cell proteomics and IL-2 and IL-15 expanded WT and Pim dKO CD8 T cell proteomics datasets is also available on the Immunological Proteome Resource website: immpres.co.uk (Brenes et al., 2023) under the Cell type(s) selection: “T cell specific” and Dataset selection: “Pim1/2 regulated TCR proteomes” and “Pim1/2 regulated IL2 or IL15 CD8 T cell proteomes”.”

As well as indicating in figure legends where proteomics datasets are first introduced in Figures 1, 2 and 4 with the text:

“An interactive version of the proteomics expression data is available for exploration on the Immunological Proteome Resource website: immpres.co.uk”

**Public Reviews:**

**Reviewer #1 (Public Review):**
Summary and Strengths:The study focuses on PIM1 and 2 in CD8 T cell activation and differentiation. These two serine/threonine kinases belong to a large network of Serine/Threonine kinases that acts following engagement of the TCR and of cytokine receptors and phosphorylates proteins that control transcriptional, translational and metabolic programs that result in effector and memory T cell differentiation. The expression of PIM1 and PIM2 is induced by the T-cell receptor and several cytokine receptors. The present study capitalized on high-resolution quantitative analysis of the proteomes and transcriptomes of Pim1/Pim2-deficient CD8 T cells to decipher how the PIM1/2 kinases control TCRdriven activation and IL-2/IL-15-driven proliferation, and differentiation into effector T cells.Quantitative mass spectrometry-based proteomics analysis of naïve OT1 CD8 T cell stimulated with their cognate peptide showed that the PIM1 protein was induced within 3 hours of TCR engagement, and its expression was sustained at least up to 24 hours. The kinetics of PIM2 expression was protracted as compared to that of PIM1. Such TCRdependent expression of PIM1/2 correlated with the analysis of both Pim1 and Pim2 mRNA. In contrast, Pim3 mRNA was only expressed at very low levels and the PIM3 protein was not detected by mass spectrometry. Therefore, PIM1 and 2 are the major PIM kinases in recently activated T cells. Pim1/Pim2 double knockout (Pim dKO) mice were generated on a B6 background and found to express a lower number of splenocytes. No difference in TCR/CD28-driven proliferation was observed between WT and Pim dKO T cells over 3 days in culture. Quantitative proteomics of >7000 proteins further revealed no substantial quantitative or qualitative differences in protein content or proteome composition. Therefore, other signaling pathways can compensate for the lack of PIM kinases downstream of TCR activation.Considering that PIM1 and PIM2 kinase expression is regulated by IL-2 and IL-15, antigen-primed CD8 T cells were expanded in IL-15 to generate memory phenotype CD8 T cells or expanded in IL-2 to generate effector cytotoxic T lymphocytes (CTL). Analysis of the survival, proliferation, proteome, and transcriptome of Pim dKO CD8 T cells kept for 6 days in IL-15 showed that PIM1 and PIM2 are dispensable to drive the IL-15mediated metabolic or differentiation programs of antigen-primed CD8 T cells. Moreover, Pim1/Pim2-deficiency had no impact on the ability of IL-2 to maintain CD8 T cell viability and proliferation. However, WT CTL downregulated the expression of CD62L whereas the Pim dKO CTL sustained higher CD62L expression. Pim dKO CTL was also smaller and less granular than WT CTL. Comparison of the proteome of day 6 IL-2 cultured WT and Pim dKO CTL showed that the latter expressed lower levels of the glucose transporters, SLC2A1 and SLC2A3, of a number of proteins involved in fatty acid and cholesterol biosynthesis, and CTL effector proteins such as granzymes, perforin, IFNg, and TNFa. Parallel transcriptomics analysis showed that the reduced expression of perforin and some granzymes correlated with a decrease in their mRNA whereas the decreased protein levels of granzymes B and A, and the glucose transporters SLC2A1 and SLC2A3 did not correspond with decreased mRNA expression. Therefore, PIM kinases are likely required for IL-2 to maximally control protein synthesis in CD8 CTL. Along that line, the translational repressor PDCD4 was increased in Pim dKO CTL and pan-PIM kinase inhibitors caused a reduction in protein synthesis rates in IL-2expanded CTL. Finally, the differences between Pim dKO and WT CTL in terms of CD62L expression resulted in Pim dKO CTL but not WT CTL retained the capacity to home to secondary lymphoid organs. In conclusion, this thorough and solid study showed that the PIM1/2 kinases shape the effector CD8 T cell proteomes rather than transcriptomes and are important mediators of IL2-signalling and CD8 T cell trafficking.Weaknesses:None identified by this reviewer.
**Reviewer #2 (Public Review):**
Summary:Using a suite of techniques (e.g., RNA seq, proteomics, and functional experiments ex vivo) this paper extensively focuses on the role of PIM1/2 kinases during CD8 T-cell activation and cytokine-driven (i.e., IL-2 or IL-15) differentiation. The authors' key finding is that PIM1/2 enhances protein synthesis in response to IL-2 stimulation, but not IL-15, in CD8+ T cells. Loss of PIM1/2 made T cells less 'effector-like', with lower granzyme and cytokine production, and a surface profile that maintained homing towards secondary lymphoid tissue. The cytokines the authors focus on are IL-15 and Il-2, which drive naïve CD8 T cells towards memory or effector states, respectively. Although PIM1/2 are upregulated in response to T-cell activation and cytokine stimulation (e.g., IL-15, and to a greater extent, IL-2), using T cells isolated from a global mouse genetic knockout background of PIM1/2, the authors find that PIM1/2 did not significantly influence T-cell activation, proliferation, or expression of anything in the proteome under anti-CD3/CD28 driven activation with/without cytokine (i.e., IL-15) stimulation ex vivo. This is perhaps somewhat surprising given PIM1/2 is upregulated, albeit to a small degree, in response to IL-15, and yet PIM1/2 did not seem to influence CD8+ T cell differentiation towards a memory state. Even more surprising is that IL-15 was previously shown to influence the metabolic programming of intestinal intraepithelial lymphocytes, suggesting cell-type specific effects from PIM kinases. What the authors went on to show, however, is that PIM1/2 KO altered CD8 T cell proteomes in response to IL-2. Using proteomics, they saw increased expression of homing receptors (i.e., L-selectin, CCR7), but reduced expression of metabolism-related proteins (e.g., GLUT1/3 & cholesterol biosynthesis) and effector-function related proteins (e.g., IFNy and granzymes). Rather neatly, by performing both RNA-seq and proteomics on the same IL2 stimulated WT vs. PIM1/2 KO cells, the authors found that changes at the proteome level were not corroborated by differences in RNA uncovering that PIM1/2 predominantly influence protein synthesis/translation. Effectively, PIM1/2 knockout reduced the differentiation of CD8+ T cells towards an effector state. In vivo adoptive transfer experiments showed that PIM1/2KO cells homed better to secondary lymphoid tissue, presumably owing to their heightened L-selectin expression (although this was not directly examined).Strengths:Overall, I think the paper is scientifically good, and I have no major qualms with the paper. The paper as it stands is solid, and while the experimental aim of this paper was quite specific/niche, it is overall a nice addition to our understanding of how serine/threonine kinases impact T cell state, tissue homing, and functionality. Of note, they hint towards a more general finding that kinases may have distinct behaviour in different T-cell subtypes/states. I particularly liked their use of matched RNA-seq and proteomics to first suggest that PIM1/2 kinases may predominantly influence translation (then going on to verify this via their protein translation experiment - although I must add this was only done using PIM kinase inhibitors, not the PIM1/2KO cells). I also liked that they used small molecule inhibitors to acutely reduce PIM1/2 activity, which corroborated some of their mouse knockout findings - this experiment helps resolve any findings resulting from potential adaptation issues from the PIM1/2 global knockout in mice but also gives it a more translational link given the potential use of PIM kinase inhibitors in the clinic. The proteomics and RNA seq dataset may be of general use to the community, particularly for analysis of IL-15 or IL-2 stimulated CD8+ T cells.

We thank the reviewer for their comments supporting the robustness and usefulness of our data.

Weaknesses:It would be good to perform some experiments in human T cells too, given the ease of e.g., the small molecule inhibitor experiment.

The suggestions to check PIM inhibitor effects in human T cell is a good one. We think an ideal experiment would be to use naïve cord blood derived CD4 and CD8 cells as a model to avoid the impact of variability in adult PBMC and to really look at what PIM kinases do as T cells first respond to antigen and cytokines. In this context there is good evidence that the signalling pathways used by antigen receptors or the cytokines IL-2 and IL-15 are not substantially different in mouse and human. We have also previously compared proteomes of mouse and human IL-2 expanded cytotoxic T cells and they are remarkably similar. As such we feel that mature mouse CD8 T cells are a genetically tractable model to use to probe the signalling pathways that control cytotoxic T cell function. To repeat the full set of experiments observed within this study with human T cells would represent 1-year of work by an experienced postdoctoral fellow.

Unfortunately, the funding for the project has come to an end and there is no capacity to complete this work.

Would also be good for the authors to include a few experiments where PIM1/2 have been transduced back into the PIM1/2 KO T cells, to see if this reverts any differences observed in response to IL-2 - although the reviewer notes that the timeline for altering primary T cells via lentivirus/CRISPR may be on the cusp of being practical such that functional experiments can be performed on day 6 after first stimulating T cells.

A rescue experiment could indeed be informative, though of course comes with challenges/caveats with re-expressing both proteins that have been deleted at once and ability to control the level of PIM kinase that is re-expressed. This work using the Pim dKO mice was performed from 2019-2021 and was seriously impacted by the work restrictions during the COVID19 pandemic. We had to curtail all mouse colonies to allow animal staff to work within the legal guidelines. We had to make choices and the Pim1/2 dKO colony was stopped because we felt we had generated very useful data from the work but could not justify continued maintenance of the colony at such a difficult time. As such we no longer have this mouse line to perform these rescue experiments.

We have however, performed a limited number of retroviral overexpression studies in WT IL-2-expanded CTL, where T cells were transfected after 24 hours activation and phenotype measured on day 6 of culture. We chose to leave these out of the initial manuscript as these were overexpression under conditions where PIM expression was already high, rather than a true test of the ability of PIM1 or PIM2 to rescue the Pim dKO phenotype. A more robust test would also have required doing these overexpression experiments in IL-15 expanded or cytokine deprived CTL where PIM kinase expression is low, however, we ran out of time and funding to complete this work.

We have provided Author response image 1 below from the experiments performed in the IL-2 CTL for interested readers. The limited experiments that were performed do support some key phenotypes observed with the Pim dKO mice or PIM inhibitors, finding that PIM1 or PIM2 overexpression was sufficient to increase S6 phosphorylation, and provided a small further increase in GzmB expression above the already very high levels in IL-2 expanded CTL.

**Author response image 1. sa3fig1:** PIM1 or PIM2 overexpression drives increased GzmB expression and S6 phosphorylation in WT IL-2 CTL. OT1 lymph node cell suspensions were activated for 24 hours with SIINFEKL peptide (10 ng/mL), IL-2 (20 ng/mL) and IL-12 (2 ng/mL) then transfected with retroviruses to drive expression of PIM1-GFP, PIM2-GFP fusion proteins or a GFP only control. T cells were split into fresh media and IL-2 daily and (A) GzmB expression and (B) S6 phosphorylation assessed by flow cytometry in GFP+ve vs GFP-ve CD8 T cells 5 days post-transfection (i.e. day 6 of culture). Histograms are representative of 2 independent experiments.

Other experiments could also look at how PIM1/2 KO influences the differentiation of T cell populations/states during ex vivo stimulation of PBMCs or in vivo infection models using (high-dimensional) flow cytometry (rather than using bulk proteomics/RNA seq which only provide an overview of all cells combined).

We did consider the idea of in vivo experiments with the Pim1/2 dKO mice but rejected this idea as the mice have lost PIM kinases in all tissues and so we would not be able to understand if any phenotype was CD8 T cell selective. To note the Pim1/2 dKO mice are smaller than normal wild type mice (discussed further below) and clearly have complex phenotypes. An ideal experiment would be to make mice with floxed Pim1 and Pim2 alleles so that one could use cre recombinase to make a T cell-specific deletion and then study the impact of this in in vivo models. We did not have the budget or ethical approval to make these mice. Moreover, this study was carried out during the COVID pandemic when all animal experiments in the UK were severely restricted. So our objective was to get a molecular understanding of the consequences of losing theses kinases for CD8 T cells focusing on using controlled in vitro systems. We felt that this would generate important data that would guide any subsequent experiments by other groups interested in these enzymes.

We do accept the comment about bulk population proteomics. Unfortunately, single cell proteomics is still not an option at this point in time. High resolution multidimensional flow cytometry is a valuable technique but is limited to looking at only a few proteins for which good antibodies exist compared to the data one gets with high resolution proteomics.

Alongside this, performing a PCA of bulk RNA seq/proteomes or Untreated vs. IL-2 vs. IL-15 of WT and PIM1/2 knockout T cells would help cement their argument in the discussion about PIM1/2 knockout cells being distinct from a memory phenotype.

We thank the reviewer for this very good suggestion. We have now included PCAs for the RNAseq and proteomics datasets of IL-2 and IL-15 expanded WT vs Pim dKO CTL in Fig S5 and added the following text to the discussion section of the manuscript (lines 429-431):

“… and PCA plots of IL-15 and IL-2 proteomics and RNAseq data show that Pim dKO IL-2 expanded CTL are still much more similar to IL-2 expanded WT CTL than to IL-15 expanded CTL (Fig S5)”.

**Recommendations for the authors:**

**Reviewer #1 (Recommendations For The Authors):**
In panel B of Figure S1, are the smaller numbers of splenocytes found in dKO fully accounted for by a reduction in the numbers of T cells or also correspond to a reduction in B cell numbers? Are the thymus and lymph nodes showing the same trend?

We’re happy to clarify on this.

Since we were focused on T cell phenotypes in the paper this is what we have plotted in this figure, however there is also a reduction in total number of B, NK and NKT cells in the Pim dKO mice (see James et al, Nat Commun, 2021 for additional subset percentages). We find that all immune subsets we have measured make up the same % of the spleen in Pim dKO vs WT mice (we show this for T cell subsets in what was formerly Fig S1C and is now Fig S1A), the total splenocyte count is just lower in the Pim dKO mice (which we show in what was formerly Fig S1B and is now Fig S1C). To note, the Pim dKO mice were smaller than their WT counterparts (though we have not formally weighed and quantified this) and we think this is likely the major factor leading to lower total splenocyte numbers.

We have not checked the thymus so can’t comment on this. We can confirm that lymph nodes from Pim dKO mice had the same number and % CD4 and CD8 T cells as in WT.

For our in vitro studies we have made sure to either use co-cultures or for single WT and Pim dKO cultures to equalise starting cell densities between wells to account for the difference in total splenocyte number. We have now clarified this point in the methods section lines 682-684

“For generation of memory-like or effector cytotoxic T lymphocytes (CTL) from mice with polyclonal T cell repertoires, LN or spleen single cell suspensions at an equal density for WT and Pim dKO cultures (~1-3 million live cells/mL)….”

**Reviewer #2 (Recommendations For The Authors):**
Line 89-99 - PIM kinase expression is elevated in T cells in autoimmunity and inhibiting therefore may make some sense if PIM is enhancing T cell activity. Why then would you use an inhibitor in cancer settings? This needs better clarification for readers, with reference to T cells, particularly given this is an important justification for looking at PIM kinases in T cells.

We thank the reviewer for highlighting the lack of clarity in our explanation here.

PIM kinase inhibitors alone are proposed as anti-tumour therapies for select cancers to block tumour growth. However so far these monotherapies haven’t been very effective in clinical trials and combination treatment options with a number of strategies are being explored. There are two lines of logic for why PIM kinase inhibitors might be a good combination with an e.g. anti-PD1 or adoptive T cell immunotherapy. (1) PIM kinase inhibition has been shown to reduce inhibitory/suppressive surface proteins (e.g. PDL1) and cytokine (e.g. TGFbeta) expression in tumour cells and macrophages in the tumour microenvironment. (2) Inhibiting glycolysis and increasing memory/stem-like phenotype has been identified as desirable for longer-lasting more potent anti-tumour T cell immunity. PIM kinase inhibition has been shown to reduce glycolytic function and increase several ‘stemness’ promoting transcription factors e.g. TCF7 in a previous study. Controlled murine cancer models have shown improvement in clearance with the combination of pan-Pim kinase inhibitors and anti-PD1/PDL1 treatments (Xin et al, Cancer Immunol Res, 2021 and Chatterjee et al, Clin Cancer Res 2019).

It is worth noting, this is seemingly contradictory with other studies of Pim kinases in T cells that have generally found Pim1/2/3 deletion or inhibition in T cells to be suppressive of their function.

We have clarified this reasoning/seeming conflict of results in the introductory text as follows (lines 90-101):

“PIM kinase inhibitors have also entered clinical trials to treat some cancers (e.g. multiple myeloma, acute myeloid leukaemia, prostate cancer), and although they have not been effective as a monotherapy, there is interest in combining these with immunotherapies. This is due to studies showing PIM inhibition reducing expression of inhibitory molecules (e.g. PD-L1) on tumour cells and macrophages in the tumour microenvironment and a reported increase of stem-like properties in PIM-deficient T cells which could potentially drive longer lasting anti-cancer responses (Chatterjee et al., 2019; Xin et al., 2021; Clements and Warfel, 2022). However, PIM kinase inhibition has also generally been shown to be inhibitory for T cell activation, proliferation and effector activities (Fox et al., 2003; Mikkers et al., 2004; Jackson et al., 2021) and use of PIM kinase inhibitors could have the side effect of diminishing the anti-tumour T cell response.”

Line 93 - The use of 'some cancers' is rather vague and unscientific - please correct phrasing like this. The same goes for lines 54 and 77 (some kinases and some analyses).

We have clarified the sentence in what is now Line 91 to include examples of some of the cancers that PIM kinase inhibitors have been explored for (see text correction in response to previous reviewer comment), which are predominantly haematological malignancies. The use of the phrase ‘some kinases’ and ‘some analyses’ in what are now Lines 52 and 75 is in our view appropriate as the subsequent sentence/(s) provide specific details on the kinases and analyses that are being referred to.

Lines 146-147 - Could it be that rather than redundancies, PIM KO is simply not influential on TCR/CD28 signalling in general but influences other pathways in the T cell?

We agree that the lack of PIM1/2 effect could also be because PIM targets downstream of TCR/CD28 are not influential and have clarified the text as follows (lines 156-161):

“These experiments quantified expression of >7000 proteins but found no substantial quantitative or qualitative differences in protein content or proteome composition in activated WT versus Pim dKO CD4 and CD8 T cells (Fig 1G-H) (Table S1). Collectively these results indicate that PIM kinases do not play an important unique role in the signalling pathways used by the TCR and CD28 to control T cell activation.”

Line 169 - Instead of specifying control - maybe put upregulate or downregulate for clarity.

We have changed the text as per reviewer suggestion (see line 183)

Line 182-183 - I would move the call out for Figure 2D to after the last call out for Figure 2C to make it more coherent for readers.

We have changed the text as per reviewer suggestion (see lines 197-200)

Line 190 - 14,000 RNA? total, unique? mRNA?

These are predominantly mRNA since a polyA enrichment was performed as part of the standard TruSeq stranded mRNA sample preparation process, however, a small number of lncRNA etc were also detected in our RNA sequencing. We left the results in as part of the overall analysis since it may be of interest to others but don’t look into it further. We do mention the existence of the non-mRNA briefly in the subsequent sentence when discussing the total number of DE RNA that were classified as protein coding vs non-coding.

We have edited this sentence as follows to more accurately reflect that the RNA being referred to is polyA+ (lines 205-207):

“The RNAseq analysis quantified ~14,000 unique polyA+ mRNA and using a cut off of >1.5 fold-change and q-value <0.05 we saw that the abundance of 381 polyA+ RNA was modified by Pim1/Pim2-deficiency (Fig 2E) (Table S2A).

Questions/points regarding figures:Figure 1 - Is PIM3 changed in expression with the knockout of PIM1/2 in mice? Although the RNA is low could there be some compensation here? The authors put a good amount of effort in to showing that mouse T cells do not exhibit differences from knocking out pim1/2 i.e., Efforts have been made to address this using activation markers and cell size, cytokines, and proliferation and proteomics of activated T cells. What do the resting T cells look like though? Although TCR signalling is not impacted, other pathways might be. Resting-state comparison may identify this.

In all experiments *Pim3* mRNA was only detected at very low levels and no PIM3 protein was detected by mass spectrometry in either wild type or PIM1/2 double KO TCR activated or cytokine expanded CD8 T cells (See Tables S1, S3, S4). There was similarly no change in Pim3 mRNA expression in RNAseq of IL-2 or IL-15 expanded CD8 T cells (See Tables S2, S6). While we have not confirmed this in resting state cells for all the conditions examined, there is no evidence that PIM3 compensates for PIM1/2 deficiency or that PIM3 is substantially expressed in T cells.

Figure 1A&B - Does PIM kinase stay elevated when removing TCR stimulus? During egress from lymph node and trafficking to infection/tumour/autoimmune site, T cells experience a period of 'rest' from T-cell activation so is PIM upregulation stabilized, or does it just coincide with activation? This could be a crucial control given the rest of the study focuses on day 6 after initial activation (which includes 4 days of 'rest' from TCR stimulation). Nice resolution on early time course though.

This is an interesting question. Unfortunately, we do not know how sensitive PIM kinases are to TCR stimulus withdrawal, as we have not tried removing the TCR stimulus during early activation and measuring PIM expression.

Based on the data in Fig 2A there is a hint that 4 hours withdrawal of peptide stimulus may be enough to lose PIM1/2 expression (after ~36 hrs of TCR activation), however, we did not include a control condition where peptide is retained within the culture. Therefore, we cannot resolve this question from the current experimental data, as this difference could also be due to a further increase in PIMs in the cytokine treated conditions rather than a reduction in expression in the no cytokine condition. This ~36-hour time point is also at a stage where T cells have become more dependent on cytokines for their sustained signalling compared to TCR stimulus.

It is worth noting that PIM kinases are thought to have fairly short mRNA and protein half lives (~5-20 min for PIM1 in primary cells, ~10 min – 1 hr for PIM2). This is consistent with previous observations that cytotoxic T cells need sustained IL-2/Jak signalling to sustain PIM kinase expression, e.g. in Rollings et al (2018) Sci Signaling, DOI:10.1126/scisignal.aap8112 . We would therefore expect that sustained signalling from some external signalling receptor whether this is TCR, costimulatory receptors or cytokines is required to drive Pim1/2 mRNA and protein expression.

Figure 1D - the CD4 WT and Pim dKO plots are identical - presumably a copying error - please correct.

We apologise for the copying error and have amended the manuscript to show the correct data. We thank the reviewer for noticing this mistake.

In Figure 1H - there is one protein found significant - would be nice to mention what this is - for example, if this is a protein that influences TCR levels this could be quite important.

The protein is Phosphoribosyl Pyrophosphate synthase 1 like 1 (Prps1l1).

This was a low confidence quantification (based on only 2 peptides) with no known function in T cells. Based on what is known, this gene is predominantly expressed in the testis (though also detected in spleen, lung, liver). A whole-body KO mouse found no difference in male fertility. No further phenotype has been reported in this mouse. See: Wang et al (2018) Mol Reprod Dev, DOI: 10.1002/mrd.23053

We have added the following text to the legend of Figure 1H to address this protein:

“Phosphoribosyl Pyrophosphate synthase 1 like 1 (Prps1l1), was found to be higher in Pim dKO CD8 T cells, but was a low confidence quantification (based on only 2 unique peptides) with no known function in T cells.”

Figure S1 - In your mouse model the reduction in CD4 T cells is quite dramatic in the spleen - is this reduced homing or reduced production of T cells through development?Could you quantify the percentage of CD45+ cells that are T cells from blood too? Would be good to have a more thorough analysis of this new mouse model.

We apologise for the lack of clarity around the Pim dKO mouse phenotype. Something we didn’t mention previously due to a lack of a formal measurement is that the Pim dKO mice were typically smaller than their WT counterparts. This is likely the main reason for total splenocytes being lower in the Pim dKO mice - every organ is smaller. It is not a phenotype reported in Pim1/2 dKO mice on an FVB background, though has been reported in the Pim1/2/3 triple KO mouse before (see Mikkers et al, Mol Cell Biol 2004 doi: 10.1128/MCB.24.13.6104-6115.2004).

The % cell type composition of the spleen is equivalent between WT and Pim dKO mice and as mentioned above, was controlled for when setting up of our in vitro cultures.

We have revised the main text and changed the order of the panels in Fig S1 to make this caveat clearer as follows (lines 138-144):

“There were normal proportions of peripheral T cells in spleens of Pim dKO mice (Fig S1A) similar to what has been reported previously in Pim dKO mice on an FVB/N genetic background (Mikkers et al., 2004), though the total number of T cells and splenocytes was lower than in age/sex matched wild-type (WT) mouse spleens (Fig S1B-C). This was not attributable to any one cell type (Fig S1A; James et al., 2021) but was instead likely the result of these mice being smaller in size, a phenotype that has previously been reported in Pim1/2/3 triple KO mice (Mikkers et al., 2004).”

Figure S1C - why are only 10-15% of the cells alive? Please refer to this experiment in the main text if you are going to include it in the supplementary figure.

With regards what was previously Fig S1C (now Fig S1A) we apologise for our confusing labelling. We were quoting these numbers as the percentage of live splenocytes (i.e. % of live cells). Typically ~80-90% of the total splenocytes were alive by the time we had processed, stained and analysed them by flow cytometry direct ex vivo. Of these CD4 and CD8 T cells made up ~%10-15 of the total live splenocytes (with most of the rest of the live cells being B cells).

We have modified the axis to say “% of splenocytes” to make it clearer that this is what we are plotting.

Figure S1 - Would be good to show that the T cells are truly deficient in PIM1/2 in your mice to be absolutely sure. You could just make a supplementary plot from your mass spec data.

This is a good suggestion and we have now included this data as supplementary figure 2.

To note, due to the Pim1 knockout mouse design this is not as simple as showing presence or absence of total PIM1 protein detection in this instance.

To elaborate: the Pim1/Pim2 whole body KO mice used in this study were originally made by Prof Anton Berns’ lab (Pim1 KO = Laird et al Nucleic Acids Res, 1993, doi: 10.1093/nar/21.20.4750, with more detail on deletion construct in te Riele, H. et al, Nature,1990, DOI: 10.1038/348649a0; Pim2 KO = Mikkers et al, Mol Cell Biol, 2004, DOI: 10.1128/MCB.24.13.6104-6115.2004). They were given to Prof Victor Tybulewicz on an FVB/N background. He then backcrossed them onto the C57BL/6 background for > 10 generations then gave them to us to intercross into Pim1/2 dKO mice on a C57BL/6 background.

The strategy for Pim1 deletion was as follows:

A neomycin cassette was recombined into the Pim1 gene in exon 4 deleting 296 Pim1 nucleotides. More specifically, the 98th pim-1 codon (counted from the ATG start site = the translational starting point for the 34 kDa isoform of PIM1) was fused in frame by two extra codons (Ser, Leu) to the 5th neo codon (pKM109-90 was used). The 3'-end of neo included a polyadenylation signal. The cassette also contains the PyF101 enhancer (from piiMo +PyF101) to ensure expression of neo on homologous recombination in ES cells.

Collectively this means that the PIM1 polypeptide is made prior to amino acid 98 of the 34 kDa isoform but not after this point. This deletes functional kinase activity in both the 34 kDa and 44 kDa PIM1 isoforms. Ablation of PIM1 kinase function using this KO was verified via kinase activity assay in Laird et al. Nucelic Acids Res 1993.

The strategy to delete Pim2 was as follows:

“For the *Pim2* targeting construct, genomic BamHI fragments encompassing *Pim2* exons 1, 2, and 3 were replaced with the hygromycin resistance gene (*Pgp*) controlled by the human *PGK* promoter.” (Mikkers et al Mol Cell Biol, 2004)

The DDA mass spectrometry data collected in Fig 1 G-H and supplementary table 1 confirmed we do not detect peptides from after amino acid residue 98 in PIM1 (though we do detect peptides prior to this deletion point) and we do not detect peptides from the PIM2 protein in the Pim dKO mice. Thus confirming that no catalytically active PIM1/PIM2 proteins were made in these mice.

We have added a supplementary figure S2 showing this and the following text (Lines 155-156):

“Proteomics analysis confirmed that no catalytically active PIM1 and PIM2 protein were made in Pim dKO mice (Fig S2).”

Figure 2A - I found the multiple arrows a little confusing - would just use arrows to indicate predicted MW of protein and stars to indicate non-specific. Why are there 3 bands/arrows for PIM2?

The arrows have now been removed. We now mention the PIM1 and PIM2 isoform sizes in the figure legend and have left the ladder markings on the blots to give an indication of protein sizes. There are 2 isoforms for PIM1 (34 and 44 kDa) in addition to the nonspecific band and 3 isoforms of PIM2 (40, 37, 34 kDa, though two of these isoform bands are fairly faint in this instance). These are all created via ribosome use of different translational start sites from a single Pim1 or Pim2 mRNA transcript.

The following text has been added to the legend of Fig 2A:

“Western blots of PIM1 (two isoforms of 44 and 34 kDa, non-specific band indicated by *), PIM2 (three isoforms of 40, 37 and 34 kDa) or pSTAT5 Y694 expression.”

Figure 2A - why are the bands so faint for PIM1/2 (almost non-existent for PIM2 under no cytokine stim) here yet the protein expression seems abundant in Figure 1B upon stim without cytokines? Is this a sensitivity issue with WB vs proteomics? My apologies if I have missed something in the methods but please explain this discrepancy if not.

There is differing sensitivity of western blotting versus proteomics, but this is not the reason for the discrepancy between the data in Fig 1B versus 2A. These differences reflect that Fig 1B and Fig 2A contrast PIM levels in two different sets of conditions and that while proteomics allows for an estimate of ‘absolute abundance’ Western blotting only shows relative expression between the conditions assessed.

To expand on this… Fig 1B proteomics looks at naïve versus 24 hr aCD3/aCD28 TCR activated T cells. The western blot data in Fig 2A looks at T cells activated for 1.5 days with SIINFEKL peptide and then washed free of the media containing the TCR stimulus and cultured with no stimulus for 4 or 24 hrs hours and contrast this with cells cultured with IL-2 or IL-15 for 4 or 24 hours. All Fig 2A can tell us is that cytokine stimuli increases and/or sustains PIM1 and PIM2 protein above the level seen in TCR activated cells which have not been cultured with cytokine for a given time period. Overexposure of the blot does reveal detectable PIM1 and PIM2 protein in the no cytokine condition after 4 hrs. Whether this is equivalent to the PIM level in the 24 hr TCR activated cells in Fig 1B is not resolvable from this experiment as we have not included a sample from a naïve or 24 hr TCR activated T cell to act as a point of reference.

Figure 4F - Your proteomics data shows substantial downregulation in proteomics data for granzymes and ifny- possibly from normalization to maximise the differences in the graph - and yet your flow suggests there are only modest differences. Can you explain why a discrepancy in proteomics and flow data - perhaps presenting in a more representative manner (e.g., protein counts)?

The heatmaps are a scaled for ‘row max’ to ‘row min’ copy number comparison on a linear scale and do indeed visually maximise differences in expression between conditions. This feature of these heatmaps is also what makes the lack of difference in GzmB and GzmA at the mRNA heatmap in Fig 5C quite notable.

We have now included bar graphs of Granzymes A and B and IFNg protein copy number in Figure 4 (see new Fig 4G-H) to make clearer the magnitude of the effect on the major effector proteins involved in CTL killing function. It is worth noting that flow cytometry histograms from what was formerly Fig 4G (now Fig 4I) are on a log-scale so the shift in fluorescence does generally correspond well with the ~1.7-2.75-fold reduction in protein expression observed.

Figure 4G - did you use isotype controls for this flow experiment? Would help convince labelling has worked - particularly for low levels of IFNy production.

We did not use isotype controls in these experiments but we are using a well validated interferon gamma antibody and very carefully colour panel/compensation controls to minimise background staining. The only ways to be 100% confident that an antibody is selective is to use an interferon gamma null T cell which we do not have. We do however know that the antibody we use gives flow cytometry data consistent with other orthogonal approaches to measure interferon gamma e.g. ELISA and mass spectrometry.

Figure 5M - why perform this with just the PIM kinase inhibitors? Can you do this readout for the WT vs. PIM1/2KO cells too? This would really support your claims for the paper about PIM influencing translation given the off-target effects of SMIs.

Regrettably we have not done this particular experiment with the Pim dKO T cells. As mentioned above, due to this work being performed predominantly during the COVID19 pandemic we ultimately had to make the difficult decision to cease colony maintenance. When work restrictions were lifted we could not ethically or economically justify resurrecting a mouse colony for what was effectively one experiment, which is why we chose to test this key biological question with small molecule inhibitors instead.

We appreciate that SMIs have off target effects and this is why we used multiple panPIM kinase inhibitors for our SMI validation experiments. While the use of 2 different inhibitors still doesn’t completely negate the concern about possible off-target effects, our conclusions re: PIM kinases and impact on proteins synthesis are not solely based on the inhibitor work but also based on the decreased protein content of the PIM1/2 dKO T cells in the IL-2 CTL, and the data quantifying reductions in levels of many proteins but not their coding mRNA in PIM1/2dKO T cells compared to controls.